# Full length RTN3 regulates turnover of tubular endoplasmic reticulum via selective autophagy

Paolo Grumati[1], Giulio Morozzi[1], Soraya Hölper[1], Muriel Mari[2], Marie-Lena IE Harwardt[3], Riqiang Yan[4], Stefan Müller[1], Fulvio Reggiori[2], Mike Heilemann[3], Ivan Dikic[1,5]*

[1]Institute of Biochemistry II, Goethe University School of Medicine, Frankfurt, Germany; [2]Department of Cell Biology, University of Groningen, University Medical Center Groningen, Groningen, Netherlands; [3]Institute of Physical and Theoretical Chemistry, Goethe University Frankfurt, Frankfurt, Germany; [4]Department of Neurosciences, Lerner Research Institute, The Cleveland Clinic Foundation, Cleveland, United States; [5]Buchmann Institute for Molecular Life Sciences, Goethe University Frankfurt, Frankfurt, Germany

**Abstract** The turnover of endoplasmic reticulum (ER) ensures the correct biological activity of its distinct domains. In mammalian cells, the ER is degraded via a selective autophagy pathway (ER-phagy), mediated by two specific receptors: FAM134B, responsible for the turnover of ER sheets and SEC62 that regulates ER recovery following stress. Here, we identified reticulon 3 (RTN3) as a specific receptor for the degradation of ER tubules. Oligomerization of the long isoform of RTN3 is sufficient to trigger fragmentation of ER tubules. The long N-terminal region of RTN3 contains several newly identified LC3-interacting regions (LIR). Binding to LC3s/GABARAPs is essential for the fragmentation of ER tubules and their delivery to lysosomes. RTN3-mediated ER-phagy requires conventional autophagy components, but is independent of FAM134B. None of the other reticulon family members have the ability to induce fragmentation of ER tubules during starvation. Therefore, we assign a unique function to RTN3 during autophagy.

*For correspondence: dikic@biochem2.uni-frankfurt.de

## Introduction

The endoplasmic reticulum (ER) is the most abundant membranous structure in the cell. It is a continuous membrane system that extends from the nuclear envelope throughout the cytosol where it forms the peripheral ER, which is a complex interconnected network of sheets, tubules and matrices (*Nixon-Abell et al., 2016*). The ER is an extremely dynamic organelle; its different subdomains are constantly remodeled in shape and total volume in order to fulfill cellular needs (*Shibata et al., 2006*; *Friedman and Voeltz, 2011*). Sheets and tubular-like networks are characterized by the presence and the collaboration of specific proteins. For example, CLIMP-63 (microtubule-binding 63 kDa cytoskeleton-linking membrane protein) localizes specifically to ER sheets (cisternae), while reticulon domain-containing proteins (RTN1 to 4) are concentrated at ER tubules. Since the reticulon domain has the ability to bend membranes, the ER shape depends on the concentration of these resident proteins and can undergo rapid structural changes in response to various environmental cues (*Voeltz et al., 2006*; *Shibata et al., 2010*). ER plasticity and the fine morphological organization of the ER support the plethora of different biological functions in which it is involved. Among others, The ER is responsible for, the biosynthesis of secretory and membrane proteins, lipids and steroids, the regulation of $Ca^{2+}$ homeostasis, and the formation of functional contact sites with other

organelles (*Borgese et al., 2006*). The ER is also the essential site for quality control of newly synthesized proteins of the secretory pathway. The ER contains two interconnected quality control pathways: the unfolded protein response (UPR) and the ER-associated protein degradation (ERAD) story. UPR activation increases the folding capacity of the ER, while the ERAD system recognizes terminally misfolded proteins and facilitates their retro-translocation into the cytoplasm where they are degraded by the ubiquitin-proteasome system (*Walter and Ron, 2011*). In this way, the ER maintains the flow of protein synthesis, folding and clearance. The ER is also actively involved in the second major degradative system of the cell: the autophagy-lysosome pathway. Autophagy is a catabolic process characterized by the formation of a double membrane structure, named autophagosome, which engulfs portions of the cytosol (like protein aggregates and organelles) and delivers them to the lysosome (*Tooze and Yoshimori, 2010*). Being the largest membranous organelle, the ER functions as the major membranes source for autophagosome formation. The nucleation of the double membrane, which is destinated to become an autophagome, takes place in specific compartments of the ER defined as omegasomes (*Biazik et al., 2015*). Interestingly, the ER has a double fate in autophagy. It initiates the process but is itself falls victim to selective autophagy, which regulats ER turnover (ER-phagy) (*Khaminets et al., 2015*). Critical determinants for selective autophagy pathways are cargo receptors, which are able to simultaneously bind the designated cargo and LC3 modifiers (*Rogov et al., 2014*; *Stolz et al., 2014*). In this manner, selective autophagy ensures the timely and specific removal of unwanted cellular components. So far, in mammals, two different ER resident proteins have been described as receptors for ER-phagy: FAM134B and SEC62 (*Khaminets et al., 2015*; *Fumagalli et al., 2016*). FAM134B is a member of the reticulon-homology-domain-containing FAM134 family and is mainly localized to the edges of the ER sheets. FAM134B binds MAP1LC3B via its LC3 interacting region (LIR) and at the same time to fragmented ER sheets and coalesces them into vesicles, which are subsequently degraded by lysosomes. Depletion of FAM134B causes an abnormal expansion of ER sheets (*Khaminets et al., 2015*). In humans, truncation mutations in the *FAM134B* gene are responsible for a severe sensory neuropathy (HSANII) (*Kurth et al., 2009*). SEC62 is a subunit of the translocon complex and functions as an autophagy receptor during recovery from ER stress. It promotes the selective clearance of excessive membrane portions to preserve proper ER structure and function (*Fumagalli et al., 2016*).

Here we identify RTN3 as a new ER-phagy receptor responsible for the selective degradation of ER tubules. An increase in the local concentration of RTN3, facilitates its oligomerization, which is sufficient to induce fragmentation of ER tubules and their subsequent lysosomal degradation in an autophagy-dependent manner. The large amino-terminal domain of RTN3, which is present in the long isoforms, contains several LIR domains and confers this, newly-identified, biological function to RTN3. Indeed, this N-terminal region is unique for each reticulon and the other members of the RTN protein family do not possess the ability to facilitate the degradation of ER tubules.

## Results

### RTN3 promotes fragmentation of ER tubules under starvation

FAM134B was the first ER-specific autophagy receptor identified. Its topology revealed a reticulon-like domain composed of a cytosolic linker region that connects two hairpin helixes (Reticulon homology domain; RHD), which anchor the protein to ER membranes, particularly to ER sheets (*Figure 1A*). The N-terminal and C-terminal domains both face the cytosolic compartment and the longer C-terminal domain presents a LIR motif responsible for the binding to MAP1LC3B and necessary to facilitate ER-phagy (*Khaminets et al., 2015*) (*Figure 1A*). Thus far, FAM134B and the subsequently identified SEC62 are the only characterized ER-phagy receptors in mammalian cells (*Khaminets et al., 2015*; *Fumagalli et al., 2016*). However, these two proteins preferentially reside in ER sheets, while the ER is divided into functionally separated structures characterized by the presence of specialized proteins (*Shibata et al., 2006*; *Friedman and Voeltz, 2011*). We, therefore, investigated if different ER-phagy receptors exist and if they are specific for other ER compartments, in particular the ER tubules. In addition to FAM134, there are other ER resident protein families containing RHDs; one of these is the reticulon family consisting of RTN1-4 (*Figure 1B*). The family structure is rather complex, due to the presence of an elevated number of splicing isoforms for each RTN (*Figure 1—figure supplement 1A*). All of the splicing products share the reticulon domain (RHD) as

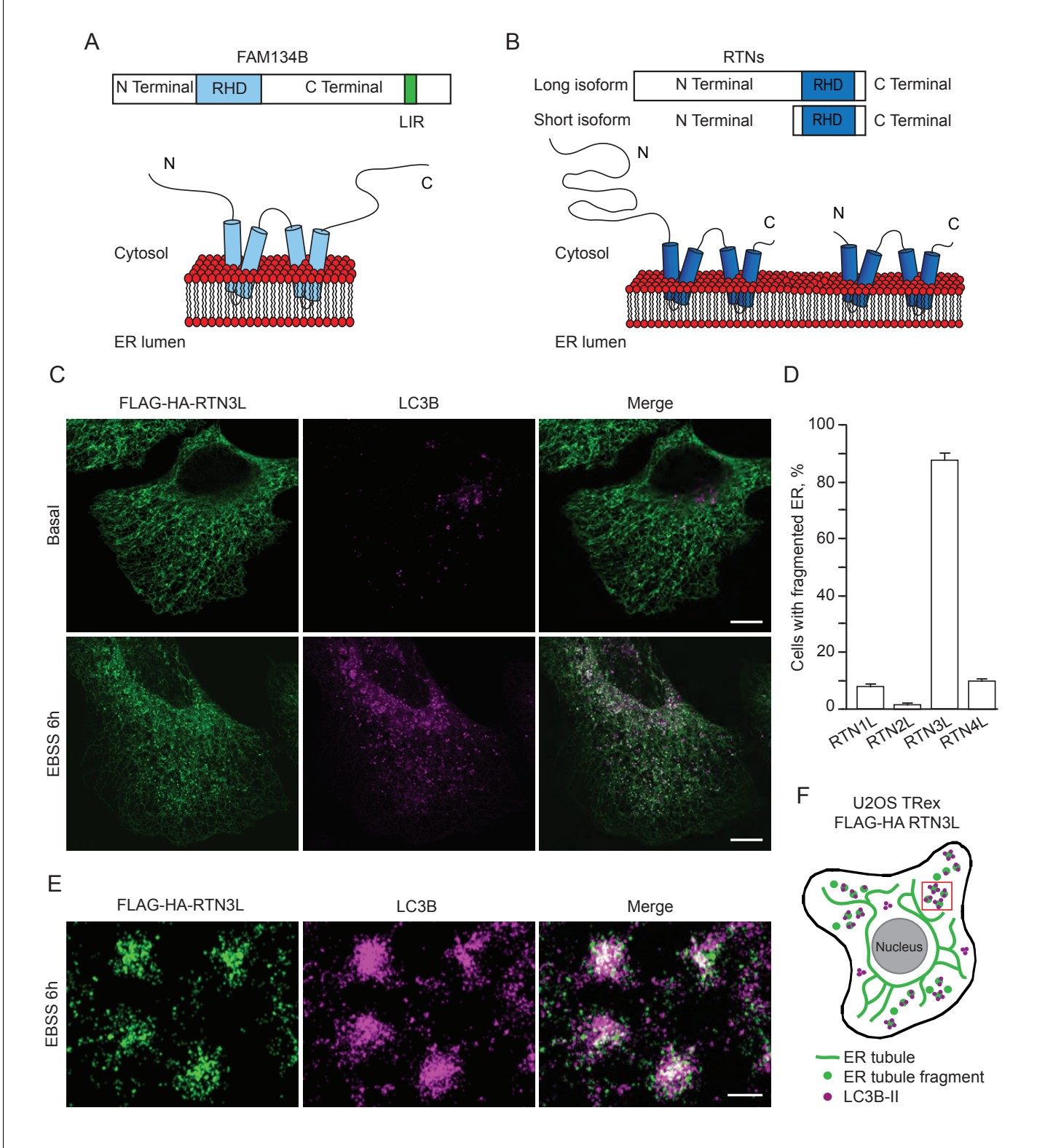

**Figure 1.** RTN3 over-expression induces ER tubules fragmentation during starvation. (**A, B**) Schematic representation of the protein structure and topology of FAM134B (**A**) and RTN3 long and short isoforms (**B**). (**C**) Immunofluorescence of HA and LC3B in U2OS TRex stable cell lines expressing FLAG-HA-RTN3L after 24 hr treatment with 1 µg/ml of doxycycline. Cells were kept in standard growing condition (DMEM with 10% FBS) or starved with EBSS for 6 hr. RTN3L was monitored using anti HA antibody, while autophagy induction was visualized using anti-LC3B antibody. Bafilomycin A1 was added at a final concentration of 200 ng/ml. Scale bars: 10 µm. (**D**) Quantification of U2OS TRex FLAG-HA-RTN1-4L cells with ER tubule fragment

*Figure 1 continued on next page*

*Figure 1 continued*

after 6 hr starvation with EBSS plus Bafilomycin A1, 200 ng/ml. Number of cells >100 for each condition. Data are representative of three independent biological replicates. Error bars indicate s.d. (**E**) Super-resolution fluorescence microscopy (*d*STORM) of ER fragments in U2OS TRex FLAG-HA-RTN3L cells stained with anti HA and anti LC3B antibodies after 6 hr starvation with EBSS plus Bafilomycin A1, 200 ng/ml. Scale bar: 0.5 μm. (**F**) Schematic representation of ER tubules fragmentation and LC3 labeling in U2OS TRex FLAG-HA-RTN3L cells. The red square indicates the level of high resolution represented in panel E.

The following figure supplements are available for figure 1:

**Figure supplement 1.** Reticulon family members.

**Figure supplement 2.** ER tubules morphology in U2OS TRex cells after over-expression of full length RTN proteins.

**Figure supplement 3.** ER tubules morphology in U2OS TRex cell after over-expression of short RTN variants.

**Figure supplement 4.** ER tubules morphology in U2OS TRex RTNs cell lines after autophagy induction via EBSS starvation.

**Figure supplement 5.** RTN3L over-expression promotes ER tubules fragmentation during autophagy induction.

well as the very short C-terminal domain, while the major variations reside in the N-terminal domain, which represents the majority of the protein within the long isoforms. We chose to analyze one long and one short isoform of each RTN and generated eight different U2OS cell lines expressing the various RTNs under the control of a doxycycline inducible promoter (*Figure 1—figure supplement 1B and C*). Twenty-four hours after induction of protein expression, we monitored whether RTN isoforms influence the morphology of ER tubules or deliver parts of them to the lysosome for degradation upon starvation (EBSS medium + Bafilomycin A1 treatment). A confocal microscopy-based screen demonstrated that all four RTNs, independently of their isoform, were able to form a defined and continuous tubular ER network (*Figure 1—figure supplements 2* and *3*). In addition to tubules, RTN2 overexpression was reported to promote the formation of several punctuate structures, mediated by its interaction with ER morphogens (*Montenegro et al., 2012*) (*Figure 1—figure supplements 2* and *3*). Upon autophagy induction, LC3B-positive puncta accumulated in the cytosol of all RTN expressing cell lines. Of note, only in the RTN3L over-expressing cells, the morphology of RTN3 decorated tubules was affected. This was evident as soon as 2 hr of starvation, but was more apparent after 4 hr and especially 6 hr. ER tubules formed by RTN3L lost their distinct morphology and progressively fragmented into smaller pieces. Strikingly, there was an almost complete co-localization between these RTN3L-decorated ER fragments and LC3B staining. Importantly, none of the other reticulons (RTN1, RTN2 and RTN4) nor the short isoform of RTN3 had either the ability to induce ER fragmentation or co-localized with LC3B puncta under conditions of starvation (*Figure 1C and D*; *Figure 1—figure supplement 4A and B*). The effective co-localization between RTN3L and LC3B was further visualized by super-resolution microscopy (*Heilemann et al., 2008*), which allowed us to focus directly on the RTN3L-decorated structures. Super-resolution images show distinct RTN3L and LC3B clusters of about 0.5 μm with a high degree of co-localization (*Figure 1E and F* and *Figure 1—figure supplement 5A*). Moreover, even at the endogenous levels, LC3B-labeled ER fragments were positive for RTN3 staining (*Figure 1—figure supplement 5B and C*). We further confirmed that the RTN3L-decorated fragments co-localize with endogenous ER protein markers CALNEXIN, BSCL2 and REEP5, under standard growth conditions and during starvation, indicating that they correspond to discrete portions of the ER. Of note, RTN3L staining always overlies with the ER tubular protein REEP5, while no co-localisation was observed with the ER sheet protein CLIMP-63 (*Figure 1—figure supplement 5D*).

Altogether, these data demonstrate that upon starvation only the long isoform of RTN3, but no other RTNs, is able to fragment ER tubules into discrete portions upon starvation and to attract LC3B, a major marker for autophagosomes.

## RTN3L clustering is required to fragment ER tubules

RTNs are known to form oligomers (*Voeltz et al., 2006*) and different isoforms of RTN3 were simultaneously found to be present in the same tissue or cell line (*Di Scala et al., 2005*). We, therefore, investigated the potential role of RTN3L clustering with itself or with RTN3S in the fragmentation of ER tubules. To this end, we used a regulated homo/hetero-dimerization system. U2OS cells were co-transfected with FRB-FLAG-RTN3L along with FKBP-HA-RTN3L or FKBP-HA-RTN3S. The FRB domain [T2098L] (modified FKBP12-rapamycin-binding domain of human mTOR kinase) is fused to the N-terminal part of RTN3L while the FKBP domain (two tandem repeats of full length human FK506 binding protein-12) is fused to the N-terminus of RTN3L and RTN3S (*Figure 2A*). Twenty-four-hour post transfection, Rapalog compound was added for 2 hr to the normal growing media (DMEM 10% FBS) in order to promote FKBP-FRB linkage (*Figure 2B*). Over-expression of the single plasmids as well as the co-transfection of FRB-FLAG-RTN3L together with FKBP-HA-RTN3L or FKBP-HA-RTN3S, in the absence of Rapalog, did not particularly affect the ER morphology except that ER membrane tubulation was promoted (*Figure 2C and D*; *Figure 2—figure supplement 1A–C*). On the contrary, the addition of Rapalog caused a significant change in ER morphology without affecting the autophagy flux (*Figure 2D*; *Figure 2—figure supplement 1B and D*). The homo-dimerization of RTN3L promoted the fragmentation of ER tubules, in a similar way to that previously observed with RTN3L over-expression during EBSS starvation (*Figure 2C and D*; *Figure 2—figure supplement 2A*). Notably, tubule fragments of the ER were labeled with LC3B 1 hr following EBSS starvation (*Figure 2E*). In contrast, the hetero-dimerization of RTN3L and RTN3S enhanced ER membrane tubulation rather than fragmentation. Indeed, ER tubules appeared longer and thicker in RTN3L and RTN3S transfected cells (*Figure 2—figure supplement 1B and C*; *Figure 2—figure supplement 2B*). Co-staining of RTN3L fragments and RTN3S large tubules with CALNEXIN, BSCL2 and REEP5 confirmed that these structures were discrete portions of ER. Of note, CLIMP-63 labeling did not co-localized with RTN3 highlighting that RTN3L fragments were exclusively portions of ER tubules and not ER sheets (*Figure 2—figure supplement 2*).

These data indicate that the oligomeric state of RTN3 modulates ER tubule morphology and its clustering, in particular, promotes ER tubules fragmentation.

## RTN3L-decorated ER fragments are delivered to lysosomes

Autophagosomal cargo is typically delivered to lysosomes and subsequently degraded (*Lamb et al., 2013*). Autophagosomes are fined structures and are therefore unable to sequester the vast ER network. Consequently, the ER architecture has to be, at least in part, disassembled in order to permit delivery of ER membrane fragments into lysosomes ([*Khaminets et al., 2015*]; *Figure 3—figure supplement 1*). To follow the fate of RTN3L-induced ER tubule fragments, we immuno-stained lysosomes with anti-LAMP1 antibody in HA-RTN3L expressing cells after 6 hr of nutrient starvation. Confocal microscopy showed that ER fragments, generated under these conditions, were localized to the interior of LAMP1-positive lysosomes (*Figure 3A*; *Figure 3—figure supplement 2A and B*; *Figure 3—figure supplement 3*). Similarly, ER fragments, obtained from the forced homo-dimerization of RTN3L, were degraded via lysosomes already after 1 hr EBSS treatment (*Figure 3—figure supplement 4A*). Super-resolution microscopy imaging of U2OS RTN3L cells, which were nutrient starved for 6 hr in the presence of Bafilomycin A1, revealed that lysosomal membranes positive for LAMP1 formed a non-homogeneous nanoscale meshwork around RTN3L and ER fragments (*Figure 3B and C*; *Figure 3—figure supplement 4B*). Next, we fused RTN3L to mCherry-EGFP double tag, which allows to follow the turnover of autophagy substrates and receptors in lysosomes, to monitor the delivery of RTN3L into lysosomes (*Kirkin et al., 2009*). In this system, lysosomal cargo delivery and degradation coincide with the appearance of mCherry-positive and EGFP-negative puncta, as the mCherry fluorescent signal is stable in the acidic milieu of lysosomes while GFP is not. HeLa cells stably expressing mCherry-EGFP-tagged RTN3L showed a clear formation of mCherry-positive, EGFP-negative puncta upon nutrient starvation (*Figure 3—figure supplement 4C*). Notably, none of the other RTNs nor RTN3S were detected inside LAMP1-positive lysosomes (*Figure 3—figure supplement 2A and B*). To further investigate the nature of the ER fragments in the autolysosomes, we performed immuno-electron-microscopy (IEM) analysis. We double labeled cryo-section from U2OS RTN3L cells, which were either grown in presence of nutrients or EBSS starved for 6 hr in EBSS supplemented with bafilomycin A1, with antibodies against the HA tag, to detect RTN3L, or

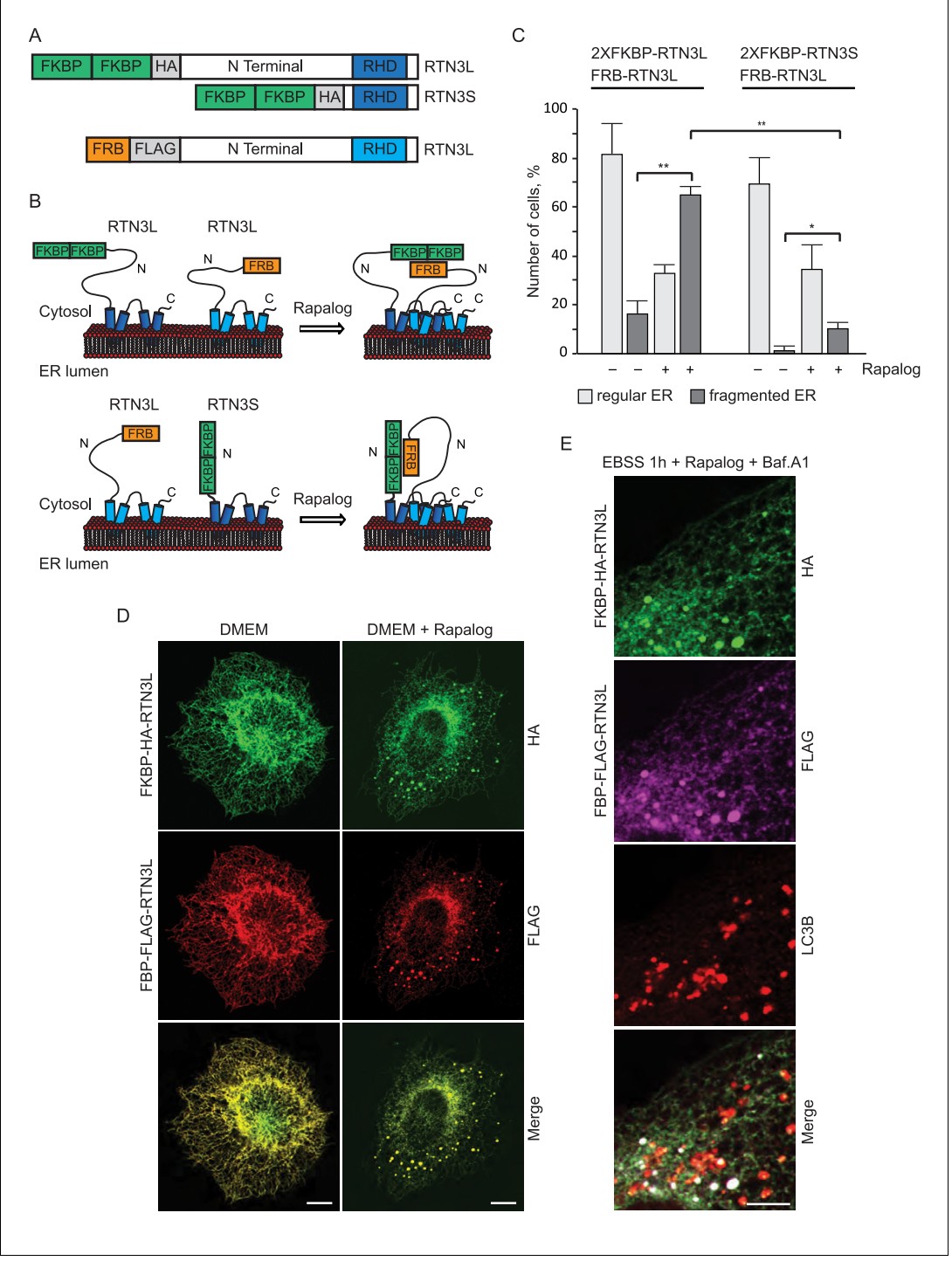

**Figure 2.** RTN3L homo-dimerization induces ER tubules fragmentation. (**A**) Schematic representation of 2XFKBP-HA-RTN3L, 2XFKBP-HA-RTN3S and FRB-FLAG-RTN3L plasmids. (**B**) Working model for the RTN3 homo/hetero-dimerization assay. (**C**) Quantification of cells presenting ER tubule fragmentation after transient co-transfection with FKBP-FKBP-HA-RTN3L/FRB-FLAG-RTN3L or 2XFKBP-HA-RTN3S/FRB-FLAG-RTN3L plasmids in standard conditions and after Rapalog treatment. Number of cells >100 for each condition. Data are representative of three independent biological replicates, *p<0.05; **p<0.01. Error bars indicate s.d. (**D,E**) U2OS TRex transiently expressing 2XFKBP-HA-RTN3L and FRB-FLAG-RTN3L. 24 hr after transfection, 500 nM Rapalog was added for 2 hr and cells were (**D**) double stained with antibodies against HA and FLAG and (**E**) triple stained with HA, FLAG and LC3B after 1 hr EBSS treatment plus 200 ng/ml Bafilomycin A1. Scale bars: 10 μm.

*Figure 2 continued on next page*

*Figure 2 continued*

The following figure supplements are available for figure 2:

**Figure supplement 1.** Homo- and hetero-dimerization of RTN3 affects ER morphology independently from autophagy.

**Figure supplement 2.** Dimerization of RTN3 modifies ER tubules structure.

the KDEL sequence, which is present in numerous ER resident proteins, and the endolysosomal protein CD63. Only after EBSS treatment, we could detect ER membranes positive for KDEL and HA-RTN3 inside autolysosomes labeled with the CD63 (*Figure 3D and E – Figure 3—figure supplement 5A and B*). Notably, we also detected RTN3L inside double-membrane, CD63 negative vesicles that represent autophagosomes (*Figure 3—figure supplement 5C*).

These findings demonstrate that RTN3L has a unique role within the RTN protein family. That is, RTN3L can drive lysosomal delivery of ER-derived tubular fragments, which provides the indication that it functions as an autophagy receptor for ER tubules.

## RTN3 absence influences selective degradation of ER tubules

In MEFs, degradation of endogenous ER proteins depends on autophagy as the starvation-induced decrease in ER protein level was impaired in *Atg5*$^{-/-}$ and *Fip200*$^{-/-}$ cells (*Figure 4A*; *Figure 4—figure supplement 1A*). Importantly, endogenous Rtn3 protein level was also regulated in an autophagy dependent manner (*Figure 4A*). To clarify the role of Rtn3 in ER turnover, we investigated how Rtn3 absence influences macro-autophagy and ER-phagy. The lack of Rtn3 affected starvation induced turnover of ER tubular proteins (*Figure 4A*; *Figure 4—figure supplement 1A*). In particular, in *Rtn3*$^{-/-}$ MEFs, degradation of Rtn1, Rtn4 and Reep5 was impaired, while ER sheet markers Climp-63 and Trap-alpha were regularly degraded as in wild-type control cells. Reconstitution of Rtn3 knockout MEFs with the full length human EGFP-RTN3L rescued nutrient deprivation-induced turnover of ER tubules markers Rtn1, Rtn4, Reep5 and EGFP-RTN3L itself was also degraded (*Figure 4B and C*; *Figure 4—figure supplement 1A*). Of interest, Fam134b degradation was not affected by the absence of Rtn3. Moreover, in *Fam134b*$^{-/-}$ MEFs we observed an opposite effect to that described for *Rtn3*$^{-/-}$ cells. EBSS treatment-induced degradation of ER tubule proteins including Rtn3, while the degradation of ER sheets proteins (Climp-63 and Trap-alpha) was blocked in *Fam134b*$^{-/-}$ MEFs (*Figure 4A*; *Figure 4—figure supplement 1A*). Rtn4 represented an exception, as reported previously by our lab, starvation-induced Rtn4 degradation was impaired in *Fam134b*$^{-/-}$ MEFs (*Figure 4A*; *Figure 4—figure supplement 1A*; [*Khaminets et al., 2015*]). Notably, Rtn3 absence did not affect macro-autophagy flux as shown by the normal Lc3b lipidation and p62 degradation upon EBSS starvation or Torin1 treatment (*Figure 4D and E*). Normal autophagosome formation and Lc3b turnover were further confirmed by transfecting *Rtn3*$^{-/-}$ cells with mCherry-EGFP-LC3B plasmid. We detected the same amount of LC3B puncta and mCherry-LC3B dots in *Rtn3*$^{-/-}$ MEFs and wild-type cells, in basal condition as well as after EBSS starvation. This indicates that the rate of LC3B degradation was equal in the two cell types (*Figure 4F–G*). Of note, the lack of Rtn3 did not affect the ER tubular networks or the general morphology of the ER (*Figure 4—figure supplement 1C and D*).

Thus, our findings in *Rtn3*$^{-/-}$ MEFs confirm the important role of this protein as a specific autophagy receptor for ER tubules. Moreover, we demonstrate that Rtn3 and Fam134b act independently, and they are ER-phagy receptors responsible for the degradation of different ER subdomains.

## Comparison of the interactomes of RTN1, RTN2, RTN3 and RTN4

To further characterize the molecular basis of selective autophagy of tubular ER as well as the unique role of RTN3L within the RTN family, we performed a MS-based screen to delineate the full interactomes of the respective long isoforms of each reticulon family members (RTN1-4). Cells expressing RTN proteins and control cells were SILAC-labeled and treated with doxycycline for 24 hr and bafilomycin A1 for 4 hr (*Figure 5—figure supplement 1A*). All four RTN proteins shared the majority of the significant interacting partners with one or more of their family members (*Figure 5A and B*;

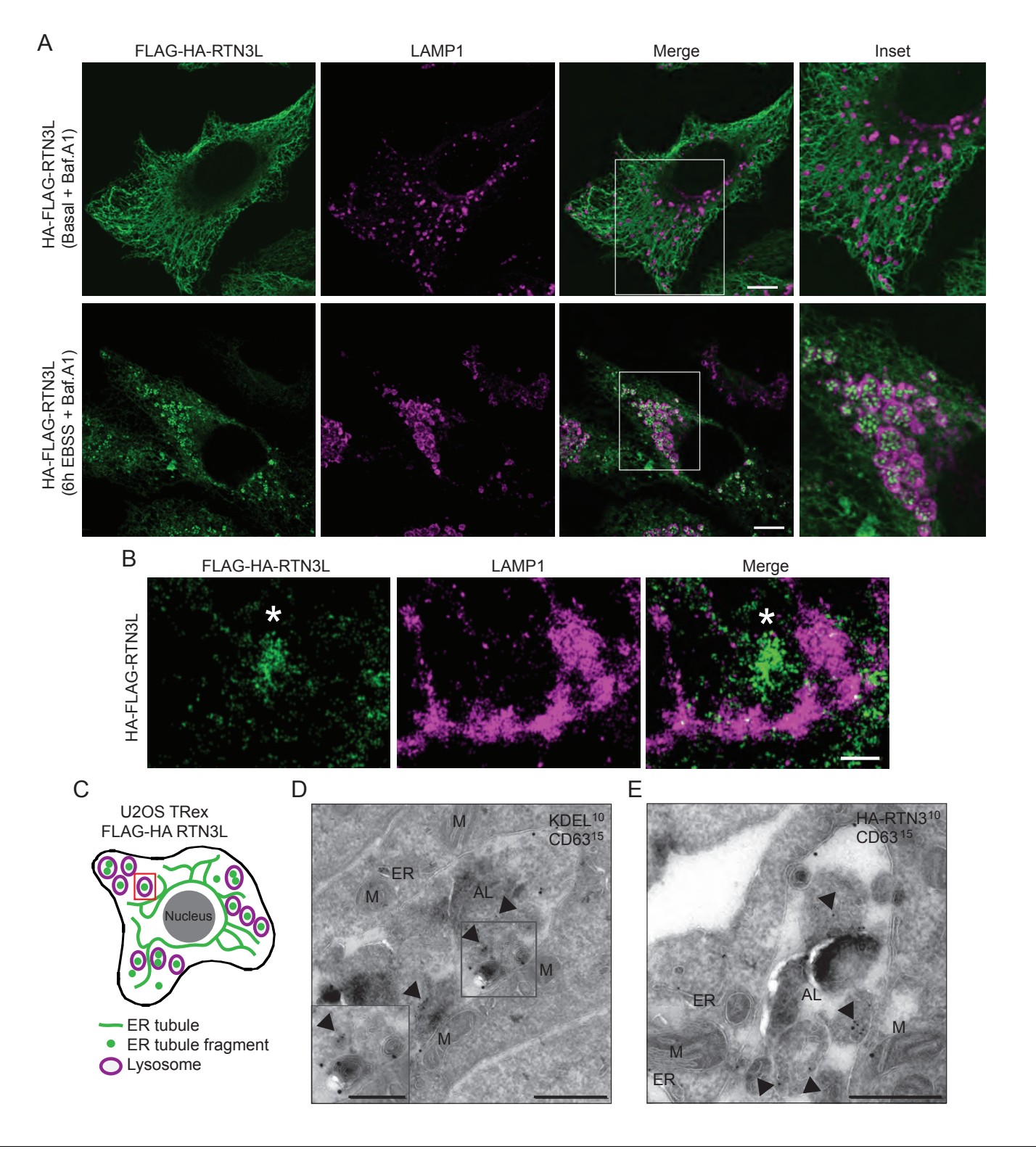

**Figure 3.** ER tubules fragments are delivered to lysosomes. (**A**) Immunofluorescence of HA and LAMP1 in U2OS TRex stable cell lines expressing FLAG-HA-RTN3L in basal growing conditions and after 6 hr starvation with EBSS plus Bafilomycin A1 200 ng/ml. RTN3L level was monitored using an anti HA antibody, while lysosomes were visualized using anti-LAMP1 antibody. Scale bars: 10 μm. (**B**) Super-resolution fluorescence microscopy (*d*STORM) of ER fragments in U2OS TRex FLAG-HA-RTN3L cells stained with anti-HA and anti-LAMP1 antibodies after 6 hr starvation with EBSS plus Bafilomycin A1, 200

*Figure 3 continued on next page*

*Figure 3 continued*

ng/ml. Asterisk indicates RTN3L positive ER tubule fragment. Scale bar: 0.5 µm. (**C**) Schematic representation of ER tubules fragmentation and their delivery to lysosome. The red square indicates the level of high resolution represented in panel B. (**D,E**) Immuno-gold labelling of cryo-sections using antibodies against CD63 (large dots, diameter 15 nm) and against either the KDEL peptide (**D**) or the HA tag (**E**) (small dots, diameter 10 nm). U2OS RTN3L cells were nutrient starved in EBSS for 6 hr in the presence of Bafilomycin A1 before being processed for IEM. AL, autolysosome; M, mitochondrion; ER, endoplasmic reticulum; Arrowheads indicates KDEL-positive ER fragments (**D**) or HA-RTN3L (**E**). Scale bar, 500 nm. Enlargement in D shows a detail of a KDEL-positive ER fragment inside an autolysosome. Scale bar, 200 nm.

The following figure supplements are available for figure 3:

**Figure supplement 1.** EBSS treatment induces fragmentation of the ER and subsequent delivery of the fragments to lysosomes.

**Figure supplement 2.** ER tubule fragments are delivered to lysosomes after autophagy induction.

**Figure supplement 3.** RTN3L fragments ER tubules and mediates their delivery to lysosomes.

**Figure supplement 4.** RTN3L is degraded via lysosomes during starvation.

**Figure supplement 5.** ER membranes and RTN3L are present in autolysosomes.

*Figure 5—figure supplement 1*; *Figure 5—figure supplement 2*; *Figure 5—source data 1* and *2*). First, we observed that the four RTN proteins had one or more peptides of the other RTNs or a peptide of their own isoform as the most enriched peptides (*Figure 5—source data 1*). The high level of interconnection among the different RTNs and their isoforms was further validated by co-immuno-precipitation after the over-expression of the respective RTNs in HEK293T cells (*Figure 5—figure supplement 3*). The short isoforms of RTN1, RTN2, RTN3 and RTN4 had a smaller number of interacting partners, but at the same time their interactors were assigned to biological functions with the highest similarity (*Figure 5—figure supplement 2*; *Figure 5—source data 2*). The long isoforms had more extended and variable interactomes although they still shared a majority of the identified partners and clusters of proteins responsible for the same biological functions. The relative number of peptides belonging to the same protein complex or signaling pathway varied amongst the RTN proteins contributing to the uniqueness of their respective interactome. Notably, the peculiarities found within the interactome were specific not only to the four unique RTN members, but were also present within isoforms of the same RTN protein. For instance, there were several examples where the amino terminal domain of the long isoforms gained new interactors that could potentially confer a unique biological function to that specific protein. RTN2L was the only reticulon that interacted with the serine palmitoyltransferase enzymatic complex. Moreover, several ubiquitin ligases and ubiquitin-related proteins like SQSTM1 and KEAP1, which are also involved in the autophagy pathway, were found to be unique interactors of RTN2L. Of note, RTN4L showed a particular affinity for ubiquitin ligases like RNF41 (*Figure 5—figure supplement 1*; *Figure 5—source data 1*). However, RTN3L was the only reticulon found to interact with the classical autophagy modifier, GABARAP-L1. Moreover, RTN3L interactors included several other unique proteins linked to the vesicular transport system and Cullin-RING-based E3 ligase complex (*Figure 5C*; *Figure 5—source data 1*). The interaction between RTN3L and GABARAP-L1 hinted at the role of this reticulon member as an autophagy receptor. Immuno-precipitation of endogenous GABARAP and GABARAP-L1, in cells overexpressing different RTN isoforms, confirmed their unique interaction with the long isoform of RTN3 (*Figure 5E*). Moreover, we further verified the interaction between RTN3 and GABARAP on endogenous level (*Figure 5F*).

Overall, our findings demonstrate that the reticulon family members share the majority of their interacting partners and all the RTN proteins preferentially interact with one another as well as with their own isoforms. However, RTN members, especially the long isoforms, also interact with their own unique set of proteins. In particular, RTN3L is the only reticulon that directly binds to the members of the LC3s/GABARAPs family. This further confirms that the long isoform of RTN3 acts as a receptor for selective autophagy of ER tubules.

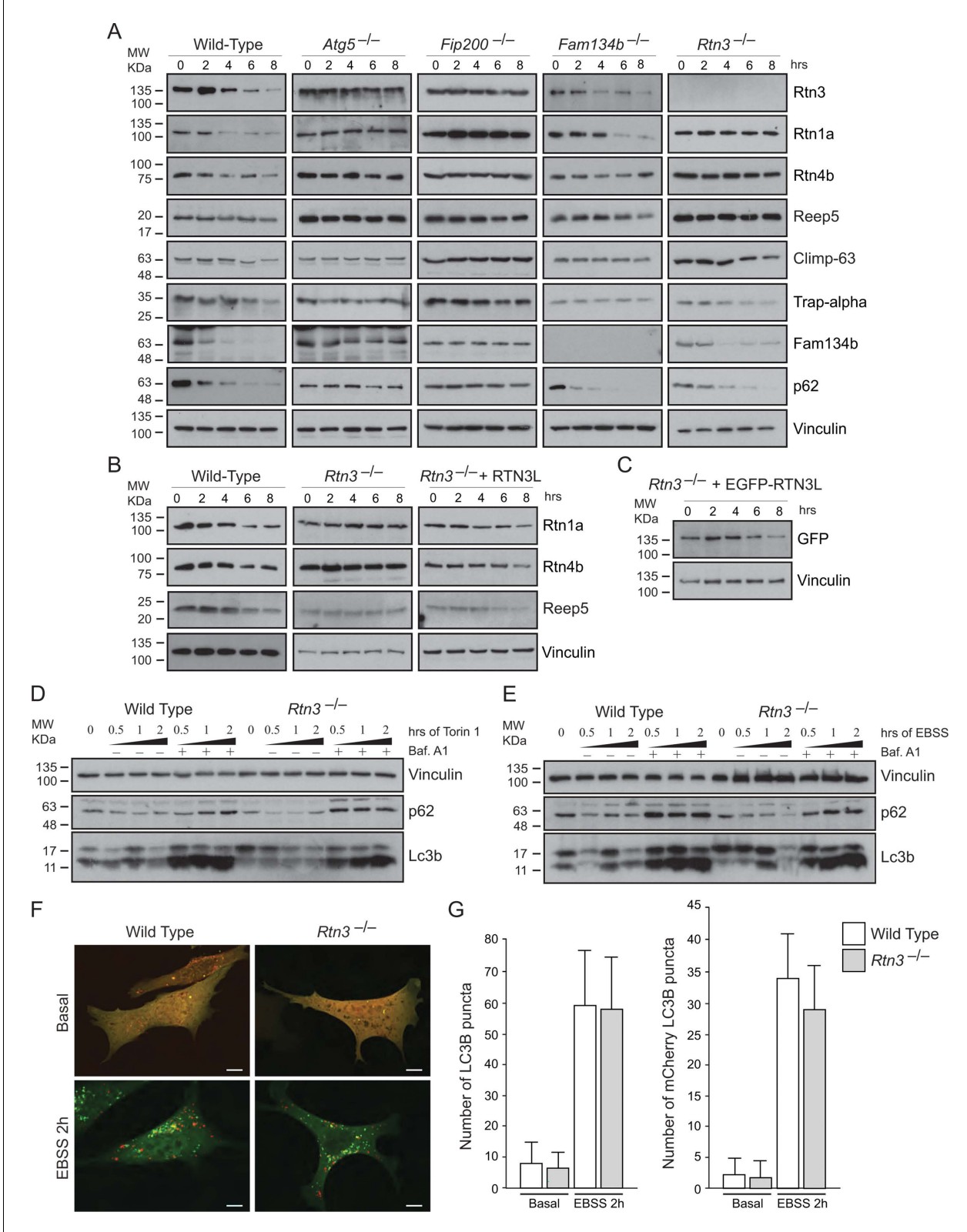

**Figure 4.** RTN3 absence impairs ER tubules turnover but not macro-autophagy flux. (**A**) Western blot analysis of ER protein turnover in wild-type, *Atg5*[−/−], *Fip200*[−/−], *Fam134b*[−/−] and *Rtn3*[−/−] MEFs. Cells were starved with EBSS for the indicate time in the presence of 100 µM cicloheximide. SQSTM1 (p62) has been used as positive control for autophagy induction. (**B**) Western blot analysis of ER tubules markers in wild-type, *Rtn3*[−/−] MEFs and *Rtn3*[−/−] MEFs reconstituted with the human EGFP-RTN3L. (**C**) Western blot for GFP in *Rtn3*[−/−] MEFs transfected with human EGFP-RTN3L. (**D,E**) Western blot

*Figure 4 continued on next page*

*Figure 4 continued*

analysis of Lc3b and p62 in wild-type and *Rtn3*⁻ᐟ⁻ MEFs. Cells were treated with 250 nM Torin1 (D) or EBSS (E) in the presence or absence of Bafilomycin A1, 200 ng/ml, for the indicated time. (F) Representative confocal imagines of wild-type and *Rtn3* knockout MEFs, transfected with mCherry-EGFP-LC3B, in standard conditions (DMEM with 10% FBS) and after 6 hr EBSS treatment. Scale bars: 10 μm. (G) Quantification of LC3B positive autophagy puncta in wild-type and *Rtn3*⁻ᐟ⁻ MEFs transfected with mCherry-EGFP-LC3B. Cells were grown in standard conditions or treated with EBSS for 2 hr. Number of cells >50 for each condition. Data are representative of three independent biological replicates. Error bars indicate s.d. No significant differences were detected between wild-type and *Rtn3*⁻ᐟ⁻ MEFs.

The following figure supplement is available for figure 4:

**Figure supplement 1.** Rtn3 absence affects ER tubules degradation but not macro-autophagy or ER morphology.

## The amino-terminal domain of RTN3 is required for binding to LC3s/GABARAPs

To further characterize the interaction of RTN3L with LC3B and GABARAP-L1, we performed in vitro pull-down experiments using GST-tagged ATG8 protein family members as affinity baits. Both endogenous RTN3 and overexpressed RTN3L were captured from cell lysates by all six LC3-like modifiers, but not by mono- or tetra-ubiquitin (*Figure 6A* and *Figure 6—figure supplement 1A*). In contrast RTN3S, which almost completely lacks the N-terminal region, did not bind any of the LC3s or GABARAPs (*Figure 6—figure supplement 1B*). Another reticulon family member, RTN2L, did not bind to LC3s/GABARAPs, despite its interaction with SQSTM1 (*Figure 6—figure supplement 1A*). To further define the interaction between RTN3L and LC3s/GABARAPs, we analyzed LC3 mutants: endogenous RTN3 did not bind when the pull-down was performed with LC3 protein family members lacking the unique amino-terminal region or LC3B with mutation in the LIR-binding pocket (*Figure 6—figure supplement 1C and D*). This indicated that RTN3L interacts with LC3 through a classical LIR motif. Bioinformatics analysis of the RTN3L protein sequence revealed the presence of six putative LIR motifs in the N-terminal region (*Figure 6B*). Substitution of the two hydrophobic amino acids, within the LIR motif, had been shown to be sufficient to ablate the interaction with ATG8 protein family members (*Rozenknop et al., 2011*). We validated the functionality of the putative LIR motifs through a sequential mutagenesis and in vitro pull down of each respective LIR mutants of RTN3L. None of the single LIR mutants, nor mutants, with only one residual LIR motif, completely lost the interaction with LC3s/GABARAPs. Complete lack of binding was obtained only after the mutagenesis of all six LIR domains (Δ6LIRs), indicating that each of the six identified LIR motifs is functional and has the ability to bind LC3 (*Figure 6C* and *Figure 6—figure supplement 1E*). A significant reduction in the fragmentation of ER tubules and ER-phagy was observed upon starvation of U2OS cells stably expressing the Δ6LIRs mutant of RTN3L as compared of WT RTN3L (*Figure 6D–F*; *Figure 6—figure supplement 2A and B*). Moreover, no lysosomal delivery of ER tubules was detected in HeLa cells stably expressing the mCherry-EGFP-RTN3LΔ6LIRs mutant, as observed by the lack of mCherry-positive puncta (*Figure 6—figure supplement 2C*). However, RTN3LΔ6LIRs was still able to tubulate ER membranes and to form an articulated ER network as well as wild-type RTN3L (*Figure 6—figure supplement 2D*). General autophagy induction and flux as visualized by LC3 lipidation and p62 degradation did not differ between U2OS RTN3L, U2OS RTN3LΔ6LIRs or U2OS TRex cells (*Figure 6—figure supplement 3*).

These data indicated that the N-terminal domain of RTN3 is required for its interaction with autophagy modifiers, and this binding is mediated by the presence of multiple LIR motifs. Moreover, functional LIR motifs are required to promote the fragmentation and subsequent delivery of ER tubules to lysosomes.

## Core autophagy machinery is required for the fragmentation of ER tubules

A comparison of the MS interactomes of RTN3L and RTN3S revealed that while 150 interactors are unique to RTN3L and 62 to RTN3S, the two isoforms share 82 p (*Figure 7A*). The common partners consisted of mainly the other RTNs family members, several ER-resident proteins, the ATP synthase complex factors and many mitochondrial proteins (*Figure 7B and C*; *Figure 7—source data 1*). The

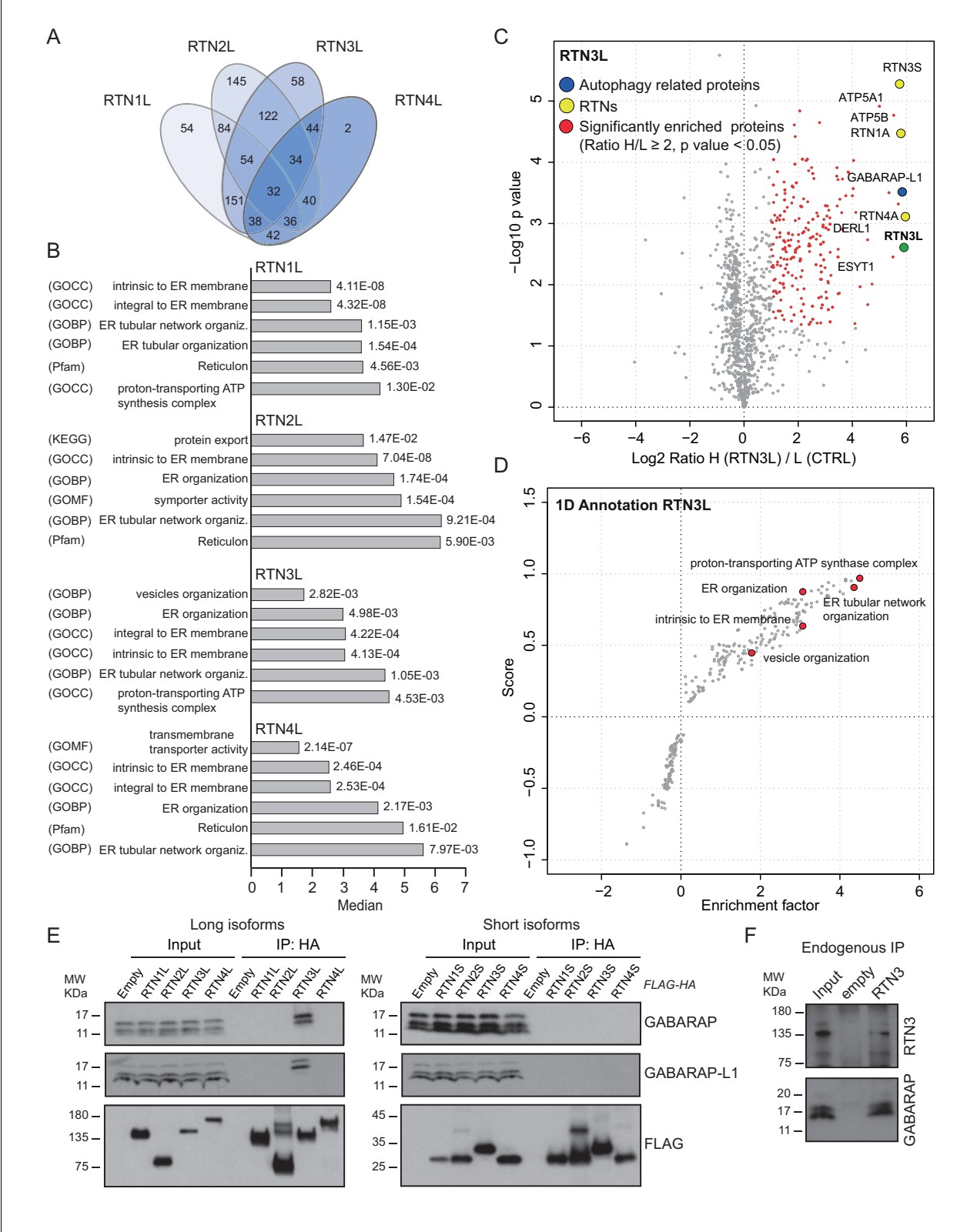

**Figure 5.** RTN3 interacts with the autophagy modifiers. (**A**) Venn diagrams of the interactors of the four RTNs. Numbers represent the identified peptides significantly enriched in three IP and mass spectrometry replicates for each RTN. (**B**) Annotation enrichment analysis of the interactors of long RTN1-4 isoforms. Bars represent the significantly enriched gene ontology biological process (GOBP), the gene ontology cellular components (GOCC), the gene ontology molecular function (GOMF), the over-expressed pathways (KEGG) and the domain enrichment (Pfam). The numeric value on the

*Figure 5 continued on next page*

Figure 5 continued

right side of the bar shows the Benjamini-Hochberg FDR value. (C) Scatter-plot for 1D annotation enrichment analysis of RTN3L interactor partners significantly enriched in three different IPs. (D) Volcano-plot for RTN3L SILAC-based interactome. Peptides with and Log2 Ratio H/L $\geq$1 and –Log10 p value > 1.3 are labeled in red. Three biological replicates were analyzed. (E) Co-IP of endogenous GABARAP and GABARAP-L1 with over-expressed long and short isoforms of RTN1-4. Over-expression was induced for 24 hr in U2OS TRex stable cell lines using 1 μg/ml of doxycycline. Bafilomycin A1 was added at the final concentration of 200 ng/ml for 2 hr. (F) Endogenous Co-IP of RTN3 with GABARAP in A549 cells. The 'empty' lane represents unconjugated beads. Bafilomycin A1, 200 ng/ml, was added for 2 hr.

The following source data and figure supplements are available for figure 5:

Source data 1. IP-interactome of RTN1, RTN2, RTN3 and RTN4 long isoforms.
Source data 2. IP-interactome of RTN1, RTN2, RTN3 and RTN4 short isoforms.
Figure supplement 1. Interactome analysis of RTN1-4L.
Figure supplement 2. Interactome analysis of RTN1-4S.
Figure supplement 3. RTN1-4 strongly interact amongst themselves.

distinct RTN3L interactors were found to reside in different organelles and function in several biological pathways (*Figure 7—source data 1*). Notably, the most enriched interactor was GABARAP-L1 (*Figure 7C*; *Figure 7—source data 1*). To further highlight the importance of the long amino-terminal domain of RTN3, we reconstituted *Rtn3*[−/−] knockout cells with the human RTN3S or RTN3LΔ6LIRs, which do not bind to the LC3 modifiers. Reconstitution of either RTN3S or RTN3LΔ6LIRs did not rescue the degradation rate of Rtn4, Rtn1 and Reep5 in *Rtn3*[−/−] MEFs. As expected, RTN3S and RTN3LΔ6LIRs protein levels remained equal after autophagy induction (*Figure 7D and E*).

To further investigate if the ability of RTN3L to fragment ER is dependent on core autophagy machinery, we knocked out the *ATG7* gene in U2OS RTN3L cells using the CRISPR-CAS9 system. Three different *ATG7* sgRNA guides as well as a GFP-CAS9 fusion protein, for subsequent cell sorting, were used to create the knock out cell line. Formation of LC3B-positive puncta was completely blocked in *ATG7*[−/−] cells. In addition, the fragmentation of ER tubules was entirely abolished in *ATG7*[−/−] cells expressing RTN3L even after 6 hr of starvation in the presence of bafilomycin A1. We also generated *FAM134B* CRISPR-CAS9 knockout cells. Contrary to the *ATG7* knockout, the absence of FAM134B did not influence the fragmentation of ER tubules mediated by RTN3L (*Figure 7F–H*).

These experiments show that RTN3L is an ER-phagy receptor that acts independently of FAM134B. However, functional core autophagy machinery is required for RTN3L-mediated fragmentation of ER membranes and their subsequent delivery to lysosomes.

## RTN3L and FAM134B act as distinct ER-phagy receptors

In order to further characterize the differences between the RTN3L and the FAM134B ER-phagy pathways, we performed a full interactome analysis in SILAC-labeled U2OS cell lines expressing FAM134B under the control of a doxycycline inducible promoter. Data analysis revealed that 179 proteins were significantly enriched with $Log_2$ (Heavy:Light [H/L]) ratios $\geq$1 and a p value $\leq$ 0.05. MAP1LC3B was the most significant interacting partner and GABARAP-L2 was also annotated as one of the most enriched proteins, thus confirming the high affinity of FAM134B for the autophagy modifiers. In addition, FAM134B strongly interacted with several other ER proteins, including ER sheet resident proteins like CLIMP-63, several components of the translocon complex as well as proteins involved in shaping ER morphology. Furthermore, FAM134B was found to interact with many elements involved in the physiological functions of the ER, such as calcium signaling and interaction with cytoskeletal microfilaments. Moreover, and intrinsic property of FAM134B, the generation of vesicles, was supported by the enrichment of candidates involved in phagocytic and endocytic vesicles formation. Interestingly, FAM134B showed a propensity to interact with proteins belonging to the ubiquitin-proteasome degradative system, as well as other proteins belonging to non-ER

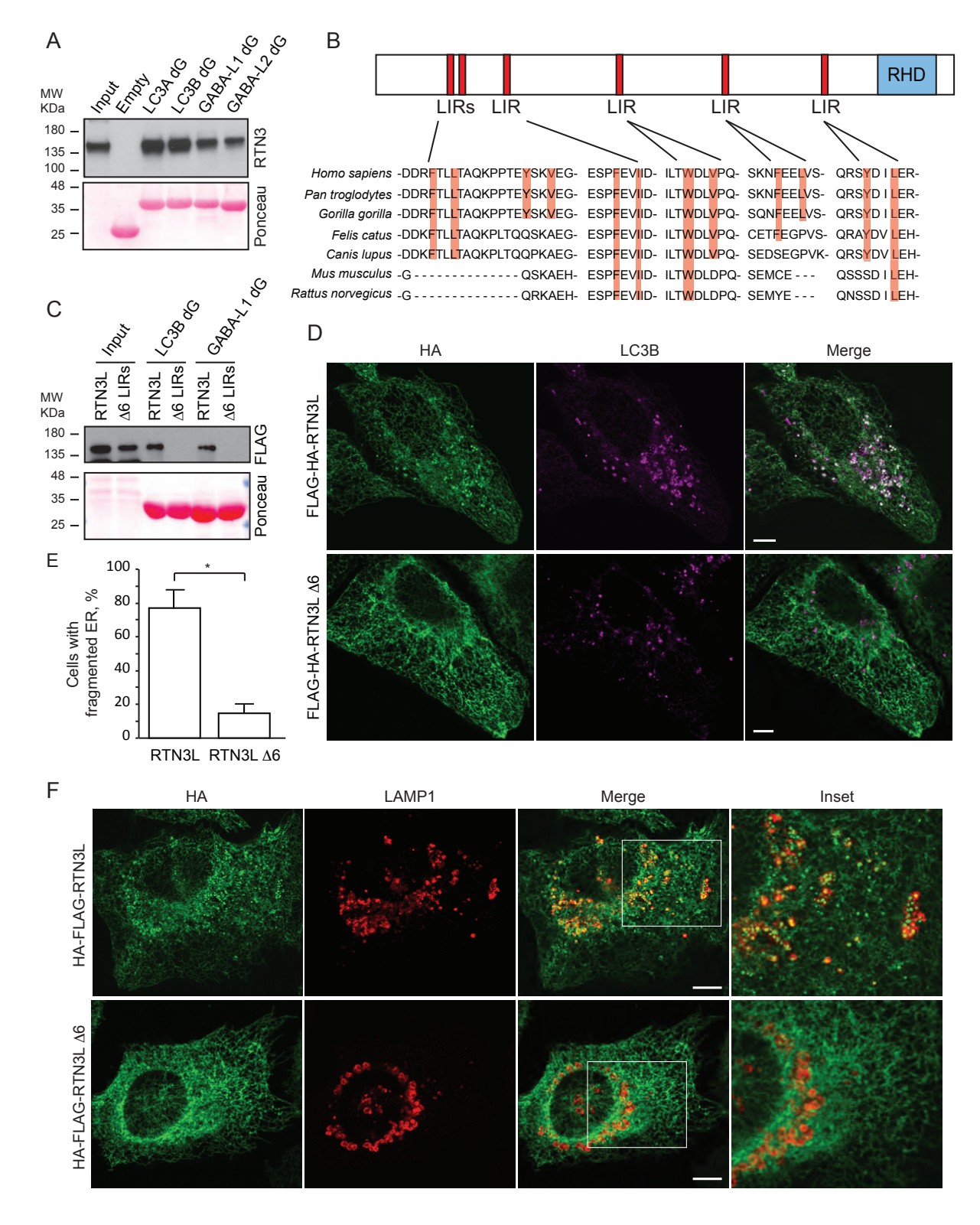

**Figure 6.** RTN3 LIR motifs are required for ER tubules fragmentation. (**A**) A549 cell lysates were added to beads with immobilized GST fusion LC3-like modifiers: GST, GST-LC3A, GST-LC3B, GST-GABARAP-L1, GST-GABARAP-L2), followed by WB using an antibody against endogenous RTN3. (**B**) Domain architecture of RTN3L and alignment of the LIR motifs. Blue: reticulon homology domain (RHD), red: LC3-interacting region (LIR). (**C**) RTN3L lacking all six LIR domains (Δ6) fails to bind to GST fusion LC3-like modifiers when over-expressed in HEK-293T cells. (**D**) Immunofluorescence of HA

*Figure 6 continued on next page*

*Figure 6 continued*

and LC3B in U2OS TRex FLAG-HA-RTN3L and FLAG-HA-RTN3LΔ6LIRs after 24 hr treatment with 1 µg/ml of doxycycline and starved for 6 hr with EBSS plus Bafilomycin A1, 200 ng/ml. RTN3L was monitored using an anti HA antibody, while autophagy induction was visualized using anti-LC3B antibody. Scale bars: 10 µm. (E) Quantification of cells presenting at least one ER tubule fragment after 6 hr starvation with EBSS plus Bafilomycin A1, 200 ng/ml. Number of cells >500 for each condition. Data are representative of three independent biological experiments. *p<0.01. Error bars represent s.d. (F) Immunofluorescence of HA and LAMP1 in U2OS TRex FLAG-HA-RTN3L or FLAG-HA-RTN3LΔ6LIRs cells induced 24 hr with 1 µg/ml of doxycycline and subsequently starved for 6 hr with EBSS plus Bafilomycin A1, 200 ng/ml. RTN3L was monitored using an anti-HA antibody, while lysosomes are visualized using anti-LAMP1 antibody. Scale bars: 10 µm.

The following figure supplements are available for figure 6:

**Figure supplement 1.** RTN3L directly binds to the LC3s/GABARAPs modifiers.

**Figure supplement 2.** LIR motifs are required to deliver RTN3L to lysosomes.

**Figure supplement 3.** RTN3L over-expression does not affect autophagy flux.

organelles like Golgi apparatus and mitochondria (*Figure 8A and B*, *Figure 8—figure supplement 1A and B*; *Figure 8—source data 1*).

To gain a better understanding of the molecular mechanisms underlying RTN3L function, we compared the RTN3L interactome with that of FAM134B. FAM134B and RTN3L shared almost half of their interactors, while 123 peptides were unique to RTN3L (*Figure 8C and D*). The most enriched interactor for both, FAM134B and RTN3L, was an autophagy modifier. While FAM134B preferentially interacted with MAP1LC3B, RTN3L favored GABARAP-L1 (*Figure 8—source data 1*). The other common interactors belonged to different complexes (ATP synthase), organelles (ER, mitochondria), and cellular compartments (cytoskeleton, vesicles) (*Figure 7E and F*). Among the unique interactors, ER-related proteins were of particular interest. RTN3L interacted mainly with the other RTNs, but it did not interact with any of the FAM134 family members (*Figure 8—figure supplement 1C*). Interestingly, FAM134B interacted preferentially with RTN2L and partially with the short RTN isoforms, but not with RTN3L (*Figure 8—figure supplement 1D–F*). Moreover, FAM134B had a stronger affinity for several proteins involved in phagocytic and endocytic vesicle formation as well as $Ca^{2+}$-related ER proteins, when compared to RTN3L (*Figure 8B*; *Figure 8—source data 1*).

These data demonstrate that RTN3L and FAM134B, two ER proteins that share a large number of potential interactors as well as a common function as autophagy receptors, do not directly interact with one another. This could be due to the special separation of these proteins in different ER subdomains, with FAM134B in sheets and RTN3L in tubules, respectively.

## Discussion

Autophagy is recognized as the principal cellular catabolic process that regulates turnover, volume, and abundance of various organelles, including the ER (*Schuck et al., 2014*; *Khaminets et al., 2016*). In this report, we identify the long isoform of the reticulon family member RTN3 (RTN3L) as a selective autophagy receptor for ER tubules. We show that RTN3L is able to interact with LC3 modifiers via six functional LIR motifs located in the long amino-terminal domain. This interaction is necessary to facilitate ER membrane fission and delivery of ER fragments to the lysosome for degradation upon autophagy induction (*Figure 9*). RTN3 itself is also degraded by the lysosome, thus acting as a *de facto* selective autophagy receptor. This property is unique to RTN3L, as none of the other reticulon family members nor the short isoform of RTN3 share this function. RTN3L loses the ability to fragment ER tubules when autophagy is completely impaired due to the loss of core machinery or when all its six of its LIR motifs are inactivated via site-specific mutagenesis. This number of functional LIR motifs is unusual for an autophagy receptor, which typically has only one LIR (*Birgisdottir et al., 2013*). An explanation could be the need for RTN3 to cluster a larger amount of LC3s modifiers and autophagic membranes, in order to fulfill its biological function as an autophagy receptor, as well as the possibility that RTN3 is itself attracted to pre-definited autophagic membranes. In both cases, this will lead to a positive loop, which determines a local concentration of

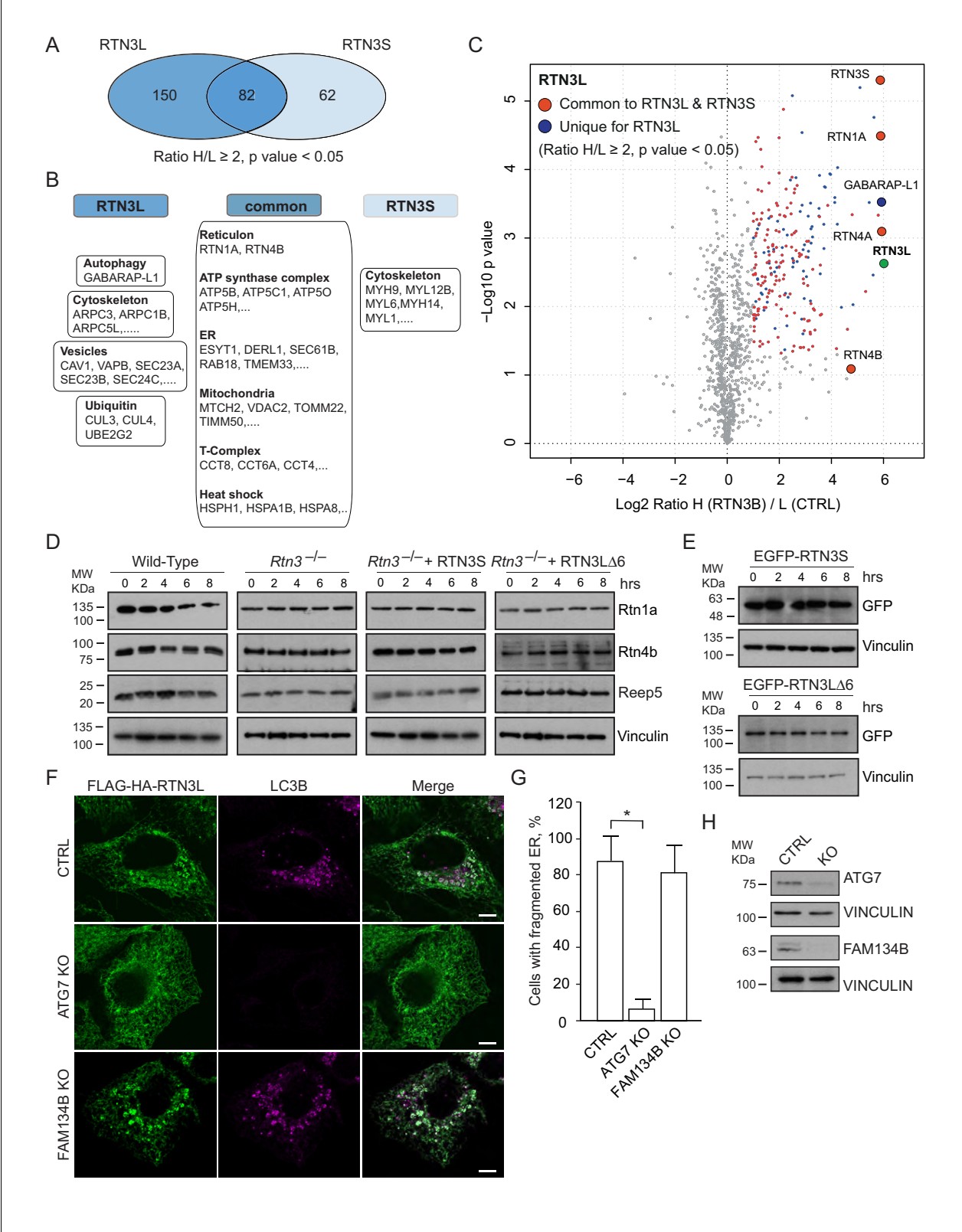

**Figure 7.** Autophagy machinery influences RTN3 ability to fragment ER tubules. (**A**) Venn diagrams of the interacting partners for RTN3L and RTNS. Numbers represent the identified proteins significantly enriched in three IP and mass spectrometry replicates for each isoform. (**B**) Schematic representation of the common and unique significantly enriched peptides for RTN3L and RTN3S. (**C**) Volcano-plot for RTN3L SILAC-based interactome. The interactors partners of RTN3L with and Log2 Ratio H/L >1 and –Log10 p value > 1.3 are labeled in dark blue. The common peptides between

*Figure 7 continued on next page*

*Figure 7 continued*

RTN3L and RTN3S, with and Log2 Ratio H/L >1 and –Log10 p value > 1.3, are labeled in red. Data represent three independent biological replicates. (D) Western blot analysis of ER tubules markers in wild-type, *Rtn3*⁻/⁻ MEFs and *Rtn3*⁻/⁻ MEFs reconstituted with the human EGFP-RTN3S or EGFP-RTN3LΔ6LIRs. (E) Western blot for GFP in *Rtn3*⁻/⁻ MEFs transfected with human EGFP-RTN3S or EGFP-RTN3LΔ6LIRs. (F) Immunofluorescence of HA and LC3B in U2OS TRex FLAG-HA-RTN3L; *FAM134B* or *ATG7* knockout cells induced for 24 hr with 1 µg/ml of doxycycline and starved for 6 hr with EBSS plus Bafilomycin A1, 200 ng/ml. RTN3L was monitored using anti HA antibody, while autophagy induction was visualized using anti-LC3B antibody. Scale bars: 10 µm. (G) Quantification of cells presenting ER tubule fragments after 6 hr starvation with EBSS plus Bafilomycin A1, 200 ng/ml. Number of cells >500 for each condition. Data are representative of three independent biological replicates. *p<0.01. Error bars indicate s.d. (H) Western blot analysis for ATG7 and FAM134B protein level in U2OS TRex FLAG-HA-RTN3L cells after *ATG7* or *FAM134B* gene knockout by CRISPR-CAS9 technology.

The following source data is available for figure 7:

**Source data 1.** Comparison of the IP-interactome of RTN3L and RTN3S.

RTN3L. On the molecular level, we showed that clustering of RTN3L is the driving force behind the fragmentation of ER tubules and their subsequent removal by ER-phagy. On the contrary, hetero-oligomerization of RTN3L with RTN3S instead favors the stabilization of ER tubules. Our observations also indicate that the autophagy machinery affects ER tubular remodeling via the co-operation between RTN3L concentration and LC3. The amino terminal domain of RTN3 plays a fundamental role in this process, and might as well have additional regulatory functions. It is a common feature for all the RTN proteins that their N-terminal regions are highly unstructured; a characteristic of many components of multi-proteins complexes with transformation ability towards alternative functions (*Mészáros et al., 2007*). The presence of multiple LIRs along the N terminal domain of RTN3 ensures that a relevant number of isoforms will contain at least one LIR motif to guarantee a proper ER turnover in all tissues. According to our model, the tubular ER network first undergoes fission events, and the resulting fragments are then engulfed by autophagosomes. During steady state tubular remodeling, the local concentration of RTN3L may increase and promote a massive recruitment of LC3 modifiers. At the same time, oligomerization of RTN3L could force the constriction of ER tubules, making them more prone to break, and the presence of LIR motifs directly targets the fragments to lysosomes. In this scenario, RTN3L is involved not only in ER tubules biogenesis, as it has been previously found to be along with the other RTNs (*Shibata et al., 2008*; *Voeltz et al., 2006*), but it also has a unique role in the control of ER tubule turnover.

We previously described how FAM134B acts as an ER-phagy receptor responsible for the homeostasis of ER sheets (*Khaminets et al., 2015*). Although both are ER membrane-bound proteins, they are localized in different ER subcompartments: FAM134B preferentially in ER sheets while RTN3 resides in tubules. Moreover, the two proteins are not closely connected: they do not interact directly and the absence of one of them does not appear to influence the biological activity of the other one. Rtn3 degradation is not affected in *Fam134b*⁻/⁻ MEFs and *vice versa*. Moreover, while Fam134b regulates the degradation of ER sheets proteins, Rtn3 absence mainly affects ER tubular protein turnover. One exception is Rtn4, which is present in both ER tubules and sheets to some extent (*GrandPré et al., 2000*), so it is subjected to the activity of both receptors. As for *Fam134b*⁻/⁻ MEFs, Rtn3 absence does not affect macro-autophagy, as well as it does not influence the ER sheets *versus* tubules ratio and ER morphology. The ER structure is complex and subdomains are characterized by the presence of key proteins involved in precise ER-shaping and biological functions (*Shibata et al., 2006*, *2010*). Therefore, it is reasonable that different resident proteins would regulate ER subdomain turnover in physiological situation or in response to a stress. Of note, SEC62, an element of the SEC61 translocon complex (*Conti et al., 2015*), has an independent function as an ER-phagy receptor during stress recovery. The peculiarity of SEC62 is its specific activation in order to resolve an ER stress situation (*Fumagalli et al., 2016*). The presence of multiple ER-phagy receptors could be an adaptive mechanism to ensure the proper homeostasis for large and complex organelles like the ER. In yeast, Atg39 and Atg40 are characterized ER-phagy receptors; Atg40 is specific for the cytosolic and cortical ER, while Atg39 facilitates turnover of perinuclear membranes (*Mochida et al., 2015*). Moreover, a similar scenario has been described for mitochondria during mitophagy (*Narendra et al., 2008*; *Lazarou et al., 2015*).

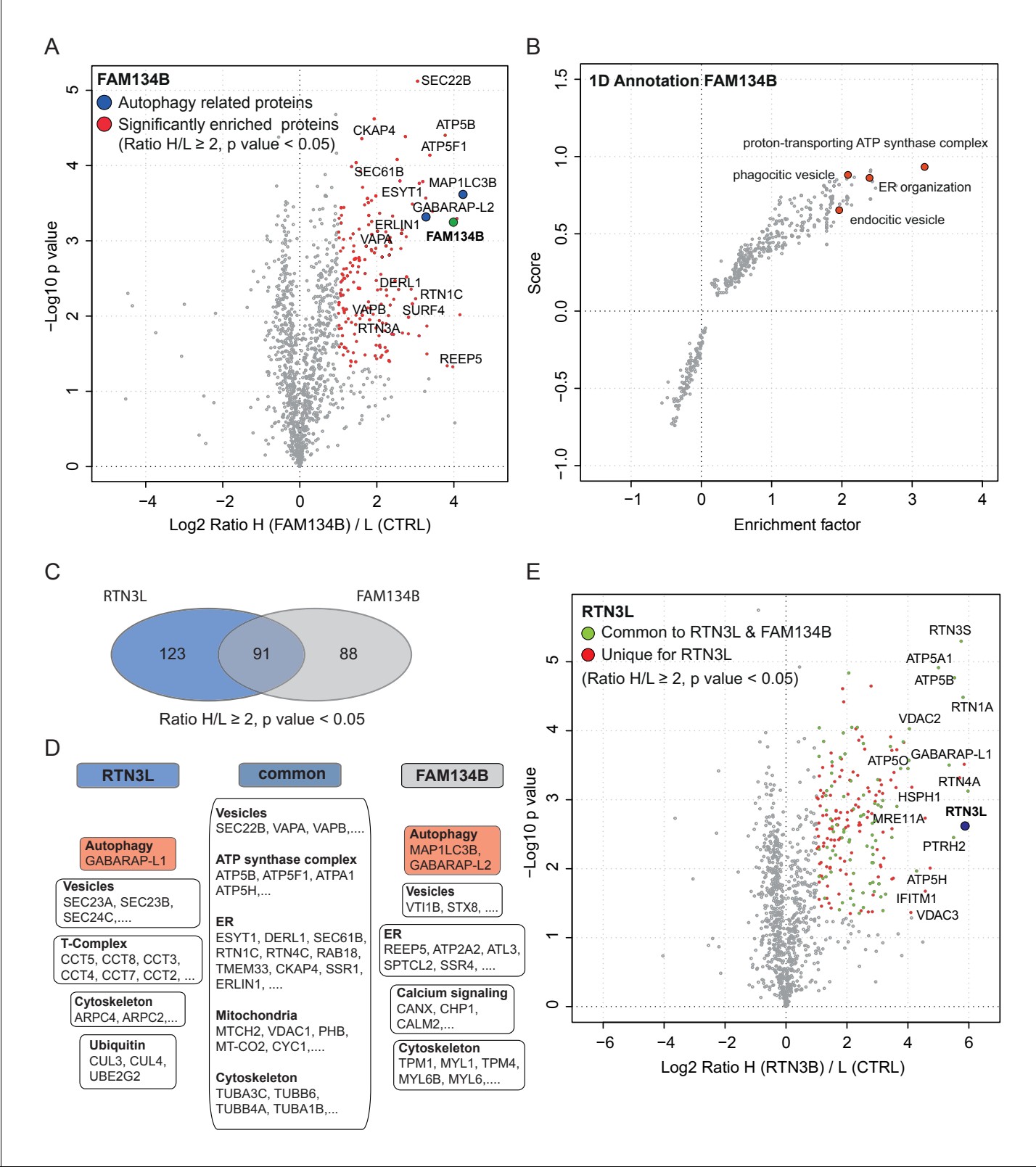

**Figure 8.** Similarities and differences between RTN3L and FAM134B. (**A**) Volcano-plot for FAM134B SILAC-based interactome. Peptides with and Log2 Ratio H/L >1 and –Log10 p value >1.3 are labeled in red. Data represent three biological replicates. (**B**) Scatter plot for 1D annotation enrichment analysis of FAM134B interacting partners significantly enriched in three different IP analyzed by mass spectrometry. (**C**) Venn diagrams of RTN3L and FAM134B interactors. Numbers represent the identified peptides significantly enriched in three IP and mass spectrometry replicates for the two baits.
*Figure 8 continued on next page*

*Figure 8 continued*

(D) Schematic representation of the common and unique interacting partners of RTN3L and FAM134B interactors (E) Volcano-plot for RTN3L SILAC-based interactome. RTN3L interactors with and Log2 Ratio H/L >1 and –Log10 p value >1.3 are labeled in dark blue. The common peptides between RTN3L and FAM134B with and Log2 Ratio H/L >1 and –Log10 p value > 1.3 are labeled in red. Data represent three biological replicates.
The following source data and figure supplement are available for figure 8:

**Source data 1.** Comparison of the IP-interactome of RTN3L and FAM134B.
**Figure supplement 1.** RTN3L and FAM134B are two independent ER-phagy receptors.

As we described, the biological function of RTN3 as an ER-phagy receptor is coupled to its unique amino terminal domain. In general, the complexity of the RTN family is mediated by the splicing rearrangements at the amino terminal that confer variable sizes to the proteins: from few amino acids to more than a thousand (*Oertle et al., 2003*; *Yang and Strittmatter, 2007*). The importance of the individual RTNs isoforms has been largely underestimated. A very limited number of scientific reports described the unique and specific functions of different isoforms for RTN4 (*GrandPré et al., 2000*; *Diekmann et al., 2005*) and RTN1 (*Iwahashi and Hamada, 2003*; *Steiner et al., 2004*). In the case of RTN3, a large number isoforms were described for both: human and murine proteins (*Di Scala et al., 2005*). Although so far only the short isoforms have been characterized and were found to be associated with several biological processes such as: apoptosis (*Kuang et al., 2005*), retrograde transport (*Wakana et al., 2005*), nuclear envelope formation (*Anderson and Hetzer, 2008*) and ER-plasma membrane contact sites (*Caldieri et al., 2017*). Our experimental settings allowed us to gain a broader over-view of the RTNs family and to 'moonlight' uncharacterized alternative functions. New proteins and complexes have now been annotated and linked to specific isoforms of RTN family members. We mainly concentrated on RTN3L, due to its interaction with GABARAP-L1 and its role in ER-phagy. However, we also noticed the interaction of SQSTM1 with RTN2, which also may be indirectly linked to selective autophagy or linked to the proteasomal system. In this regard, the long isoforms of RTN2, RTN3 and RTN4 have an affinity for the ubiquitin ligases and ubiquitin-related proteins, which may directly influence their function. For example, ubiquitination of RTN4B induces structural re-arrangements of ER tubules upon *Legionella* infection (*Kotewicz et al., 2017*), via a novel phosphoribose-linked ubiquitin modification (*Bhogaraju et al., 2016*). Ubiquitination, as well as other post-translational modifications, could potentially act as signals to induce RTN3 redistribution along the ER tubules in order to promote specific biological functions, like its homo-dimerization.

In mice, the complete lack of Rtn3 did not reveal any relevant phenotype (*Shi et al., 2014*) and there are not human pathologies genetically linked to RTN3. Although there are a few signs of RTN3S accumulation in dystrophic neurites from Alzheimer's patients exist (*Hu et al., 2007*). Nevertheless, we have now identified a new role for RTN3, as an ER-phagy receptor. This function is mediated by its amino terminal domain. These findings pave the way to further investigate RTN3 biology with a particular attention to the tissues and cells where the long isoforms are expressed and the ER tubular network is subjected to intense remodeling and turnover.

## Materials and methods

### Cloning procedures and DNA mutagenesis

cDNAs were cloned into pDONR223 vector using the BP Clonase Reaction Kit (Invitrogen, Carlsbad, CA, USA) and further recombined into GATEWAY destination vectors: iTAP MSCV-N-FLAG-HA IRES-PURO, pHAGE-N-eGFP, pcDNA3.1-N-FLAG, pcDNA5-FRT/TO-N-mCherry-EGFP though the LR Clonase Reaction Kit (Invitrogen). Mutations were generated via site-directed mutagenesis according to standard protocols. The pcDNA5-FRT/TO-N-mCherry-EGFP plasmid was obtained from pcDNA5-FRT/TO-N-SGFP after digestion with HINDIII and NotI and subsequent ligation with mCherry-EGFP segment. FLAG-RTN3L was cloned into pC4-RhE-FRB(T2098L) plasmid using *Spe*I and *Bam*HI as unique restriction sites. HA-RTN3L and HA-RTN3S were cloned into pC4M-F2E-FKBP

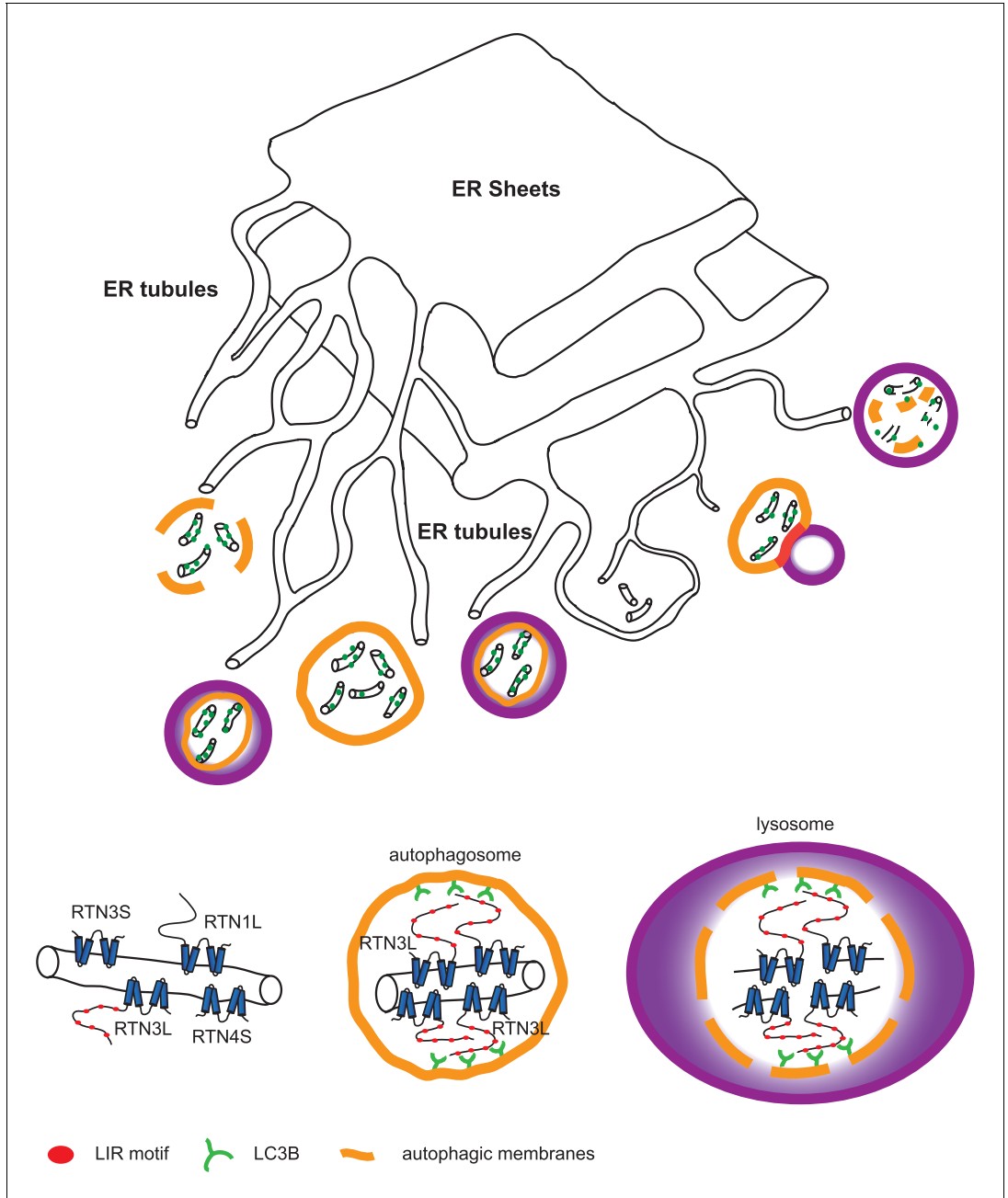

**Figure 9.** Model for ER tubules degradation. To drive ER tubules turnover, local RTN3L homo-dimerizes and leads to the recruitment of autophagic membranes and ER tubules fragmentation. Autophagosomes containing ER tubules as cargo, subsequently fuse with lysosomes.

plasmid using SpeI and BamHI as unique restriction sites. N-myr signal was removed from the plasmids through site-direct mutagenesis. All cDNAs used in the manuscript are reported in *Table 1*.

## Cell culture

HEK293T (RRID: CVCL_0063) and A549 (RRID: CVCL_0023) were provided from the American Type Culture Collection (Manassas, VA). Their identities were authenticated by STR analysis. U2OS TRex cells were provided by Prof. Stephen Blacklow (Brigham and Women's Hospital and Harvard Medical School) (*Gordon et al., 2009*), HeLa TRex were provided by Prof. S. Taylor (Manchester University), *Atg5⁻/⁻* MEFs were provided by Prof. Noboru Mizushima (Tokyo University)(*Kuma et al., 2004*),

**Table 1.** Plasmids and cDNAs related to the experimental procedures. Plasmids used in the manuscript are listed below.

| Plasmid/epitope-tag | Gene/Mutation | Reference |
|---|---|---|
| pGEX-4T1 alone | GST only | (*Kirkin et al., 2009*) |
| pGEX-4T1 LC3A dG | Deletion of terminal glycine | (*Kirkin et al., 2009*) |
| pGEX-4T1 LC3B dG | Deletion of terminal glycine | (*Kirkin et al., 2009*) |
| pGEX-4T1 LC3C dG | Deletion of terminal glycine | (*Kirkin et al., 2009*) |
| pGEX-4T1 GABARAP dG | Deletion of terminal glycine | (*Kirkin et al., 2009*) |
| pGEX-4T1 GABARAP-L1 dG | Deletion of terminal glycine | (*Kirkin et al., 2009*) |
| pGEX-4T1 GABARAP-L2 dG | Deletion of terminal glycine | (*Kirkin et al., 2009*) |
| pGEX-4T1 Ub | Human Ubiquitin | (*Kirkin et al., 2009*) |
| pGEX-4T1 4XUb | Human 4 X linear Ubiquitin | (*Kirkin et al., 2009*) |
| pGEX-4T1 dN LC3A dG | Lacking unique N-terminus and deletion of terminal glycine | This study |
| pGEX-4T1 dN LC3B dG | Lacking unique N-terminus and deletion of terminal glycine | This study |
| pGEX-4T1 dN GABARAP-L1 dG | Lacking unique N-terminus and deletion of terminal glycine | This study |
| pGEX-4T1 dN GABARAP-L2 dG | Lacking unique N-terminus and deletion of terminal glycine | This study |
| pGEX-4T1 LC3B F52A-V53A dG | Mutated LIR binding pocket | This study |
| RTN1A cDNA | Human RTN1A, long isoform (NM_021136.2) | Open Biosystem (BC090862) |
| pCMV-SPORT6-RTN2A | Human RTN2A, long isoform (NM_005619.4) | Source Bioscience (IRATp970B0996D) |
| pReceiver-M06-RTN3B | Human RTN3B, long isoform (NM_201428.2) | Genocopoeia, TebuBio (EX-Z3044-M06) |
| pcDNA3.1-RTN4A-myc | Human RTN4A, long isoform (NM_020532.4) | Provided by S. Strittmatter (*GrandPré et al., 2000*) |
| pcDNA3.1-RTN4C-myc | Human RTN4C, short isoform (NM_007008.2) | Provided by S. Strittmatter (*GrandPré et al., 2000*) |
| RTN3A cDNA | Human RTN3A, short isoform (NM_006054.3) | Open Biosystem (BC011394) |
| pDONR223-RTN1L | Human RTN1A, long isoform | This study |
| pDONR223-RTN2L | Human RTN2A, long isoform | This study |
| pDONR223-RTN3L | Human RTN3B, long isoform | This study |
| pDONR223-RTN4L | Human RTN4A, long isoform | This study |
| pDONR223-RTN1S | Human RTN1C, short isoform | This study |
| pDONR223-RTN2S | Human RTN2C, short isoform | This study |
| pDONR223-RTN3S | Human RTN3A, short isoform | This study |
| pDONR223-RTN4S | Human RTN4C, short isoform | This study |
| iTAP-FLAG-HA-RTN1L | Human RTN1A, long isoform | This study |
| iTAP-FLAG-HA-RTN2L | Human RTN2A, long isoform | This study |
| iTAP-FLAG-HA-RTN3L | Human RTN3B, long isoform | This study |
| iTAP-FLAG-HA-RTN4L | Human RTN4A, long isoform | This study |
| iTAP-FLAG-HA-RTN1S | Human RTN1C, short isoform | This study |
| iTAP-FLAG-HA-RTN2S | Human RTN2C, short isoform | This study |
| iTAP-FLAG-HA-RTN3S | Human RTN3A, short isoform | This study |
| iTAP-FLAG-HA-RTN4S | Human RTN4C, short isoform | This study |
| pHAGE-EGFP-RTN1L | Human RTN1A, long isoform | This study |
| pHAGE-EGFP-RTN2L | Human RTN2A, long isoform | This study |

*Table 1 continued on next page*

*Table 1 continued*

| Plasmid/epitope-tag | Gene/Mutation | Reference |
|---|---|---|
| pHAGE-EGFP-RTN3L | Human RTN3B, long isoform | This study |
| pHAGE-EGFP-RTN4L | Human RTN4A, long isoform | This study |
| pHAGE-EGFP-RTN3S | Human RTN3A, short isoform | This study |
| pHAGE-EGFP-RTN3LΔ6LIRs | Human RTN3B, long isoform all mutant LIRs | This study |
| pcDNA3.1-FLAG-RTN3L | Human RTN3B, long isoform | This study |
| pcDNA3.1-FLAG-RTN3L LIR mutant ΔFTLL | Human RTN3B, long isoform DDRFTLLTA/DDRATLATA aa202-210 | This study |
| pcDNA3.1-FLAG-RTN3L LIR mutant ΔYSKV | Human RTN3B, long isoform PTEYSKVEG/PTEASKAEG aa214-222 | This study |
| pcDNA3.1-FLAG-RTN3L LIR mutant ΔFEVI | Human RTN3B, long isoform ESPFEVIID/ESPAEVAID aa245-253 | This study |
| pcDNA3.1-FLAG-RTN3L LIR mutant ΔWDLV | Human RTN3B, long isoform ILTWDLVPQ/ILTADLAPQ aa339-347 | This study |
| pcDNA3.1-FLAG-RTN3L LIR mutant ΔFEEL | Human RTN3B, long isoform SKNFEELVS/SKNAEEAVS aa552-561 | This study |
| pcDNA3.1-FLAG-RTN3L LIR mutant ΔYDIL | Human RTN3B, long isoform QRSYDILER/QRSADIAER aa787-795 | This study |
| pcDNA3.1-FLAG-RTN3LΔ5LIRs-WDLV | Human RTN3B, long isoform all mutant LIRs except WDLV | This study |
| pcDNA3.1-FLAG-RTN3LΔ5LIRs-YSKV | Human RTN3B, long isoform all mutant LIRs except YSKV | This study |
| pcDNA3.1-FLAG-RTN3LΔ5LIRs-YDIL | Human RTN3B, long isoform all mutant LIRs except YDIL | This study |
| pcDNA3.1-FLAG-RTN3LΔ5LIRs-FTLL | Human RTN3B, long isoform all mutant LIRs except FTLL | This study |
| pcDNA3.1-FLAG-RTN3LΔ5LIRs-FEEL | Human RTN3B, long isoform all mutant LIRs except FEEL | This study |
| pcDNA3.1-FLAG-RTN3LΔ5LIRs-FEVI | Human RTN3B, long isoform all mutant LIRs except FEVI | This study |
| pcDNA3.1-FLAG-RTN3LΔ6LIRs | Human RTN3B, long isoform all mutant LIRs | This study |
| pDONR223-RTN3LΔ6LIRs | Human RTN3B, long isoform all mutant LIRs | This study |
| iTAP-FLAG-HA-RTN3LΔ6 LIRs | Human RTN3B, long isoform all mutant LIRs | This study |
| pcDNA5 FRT-TO mCherry-EGFP-RTN3L | Human RTN3B, long isoform | This study |
| pcDNA5 FRT-TO mCherry-EGFP-RTN3LΔ6LIRs | Human RTN3B, long isoform all mutant LIRs | This study |
| pDONR223-FAM134A | Human FAM134A | This study |
| pDONR223-FAM134B | Human FAM134B | (*Khaminets et al., 2015*) |
| pDONR223-FAM134C | Human FAM134C | This study |
| iTAP-FLAG-HA-FAM134A | Human FAM134A | This study |
| iTAP-FLAG-HA-FAM134B | Human FAM134B | (*Khaminets et al., 2015*) |
| iTAP-FLAG-HA-FAM134C | Human FAM134C | This study |
| pC4-RhE FRB (T2098L) | | Provided by R. Youle |
| pC4M-F2E FKBP | | Provided by R. Youle |
| pC4-RhE FRB (T2098L)-FLAG-RTN3L | Human RTN3B, long isoform | This study |
| pC4M-F2E FKBP-HA-RTN3L | Human RTN3B, long isoform | This study |
| pC4M-F2E FKBP-HA-RTN3S | Human RTN3A, short isoform | This study |
| mCherry-EGFP LC3B | Human LC3B | (*Khaminets et al., 2015*) |
| GFP-SEC63 | Human SEC63 | Provided by H. Farhan |

*Fam134b⁻/⁻* MEFs were provided by Prof. Christian Hübner (Jena University) (*Khaminets et al., 2015*), *Rtn3⁻/⁻* were provided by Prof. Riqiang Yan (Lerner Research Institute, The Cleveland Clinic Foundation), *Fip200⁻/⁻* MEFs were provided by Prof. Jun-Lin Guan (Cincinnati University) (*Gan et al., 2006*). All cell lines were regularly tested negative for the presence of mycoplasma using LookOut Mycoplasma PCR Detection Kit (Sigma-Aldrich [Merck], Darmstadt, Germany). Cells were maintained at 37°C with 5% $CO_2$ in DMEM medium (Gibco, Paisley, UK) supplemented with 10% fetal bovine serum (Gibco) and 100 U/ml penicillin and streptomycin (Gibco). Starvation was conducted by incubating cells in EBSS medium (Gibco). Cells were treated with 1 µg/ml of doxycycline (Sigma-Aldrich), 200 ng/ml bafilomycin A1 (LC-Laboratories, Woburn, MA, USA), 100 µM cycloheximide (AppliChem PanReac, Darmstadt, Germany), 250 nM Torin1 (LC-Laboratories), 500 ng Rapalog (Clontech [Takara Bio USA, Inc. Mountain View, CA]) for the indicated time periods. For each treatment cells were plated the day before in order to perform the experiments when cells had a final confluency of 50–60%. For transient expression, DNA plasmids were transfected with GeneJuice (Merck-Millipore #70967-4, Darmstadt, Germany) or Turbofect (Thermo Scientific #R0531, Waltham, MA, USA) according to the instructions of the manufacturers.

## Generation and propagation of stable and inducible cell lines

HeLa TRex and U2OS TRex cell lines were used to generate stable cell lines using Flp-IN T-Rex system (Invitrogen) or MSCV iTAP N-FLAG-HA retroviral vector. Induction was performed with 1 µg/ml doxycycline for the indicated time. Stable cell lines were produced using a retroviral or lentiviral virus infection. Viruses were produced using HEK 293T cells. U2OS TRex stable and inducible cell lines were generated cloning the cDNAs into MSCV iTAP N-FLAG-HA retroviral vector. U2OS TRex ATG7 and FAM134B KO cell lines were created using the CRISPR-CAS9 technology. The HeLa TRex cell line for creating stable cell lines using the Flp-In TM System was kindly provided by S. Taylor (University of Manchester). mCherry-EGFP-RTN3B, mCherry-EGFP-RTN3BΔ6LIRs mutant, constructs were cloned into pcDNA5/FRT/TO vector using the GATEWAY technology and transfected with recombinase pOG44 into Flp-In HeLa TRex cells. Hygromycin-resistant cells were expanded. All stable cell lines were continuously maintained in their respective selection-containing media. *Rtn3⁻/⁻* MEFs were generated from Rtn3 knockout mice (*Shi et al., 2014*). Primary cells were isolated from E12.5 embryos and immortalized using SV40 large T antigen. Immortalized Rtn3 knockout MEFs were reconstituted with human EGFP-RTN3B; EGFP-RTN3BΔ6LIRs; EGFP-RTN3S and FACS sorted in order to enrich for EGFP positive cells. Plasmids transfection was performed using GenJet reagent (SignaGen Laboratories #SL100489-MEF, Rockville, MD, USA).

## Antibodies used for western blot, immuno-precipitation and immuno-fluorescence staining

The complete list of primary antibodies used for the experiments is reported in *Table 2*. Secondary antibody, HRP-conjugated, were purchased from Santa Cruz Biotechnology (Dallas, TX, USA). For immuno-fluorescence staining, the following antibodies were used: anti-rabbit Alexa 555 (Life Technology; A31572; Ober-Olm, Germany), anti-rabbit Alexa 647 (Life Technology; A21244), anti-rat Cy3 (Jackson Lab; 712166153; Bar Harbor, MN, USA), anti-rat Alexa 647 (Life Technology; A21247), anti-mouse Alexa 647 (Life Technology; A31626), anti-mouse Alexa 532 (Life Technology; A11002).

## Generation of knockout cell lines using the CRISPR-CAS9 lentiviral system

Specific RNA-guides were designed using the GPP Web Portal of the Broad Institute (http://www.broadinstitute.org/rnai/public/analysis-tools/sgrna-design). The complete list of sgRNA guides used for the experiments is reported in *Table 3*. Oligos were annealed and phosphorylated following the protocol by (*Ran et al., 2013*) and cloned into the Lentiviral vectors containing the CAS9. pLenti-Puro, pLenti-Puro-EGFP or pLenti-Neo were cut with BsmBI enzyme and, after gel purification, 100 ng of the cut vector was used for ligation with 4 µl of the annealed oligos (previously diluted 1:200). Ligation was performed with T4 ligase for 1 hr at room temperature. The ligation product was then transformed using XL1-Blue competent cells. To generate a knockout cell line, we infected our target cells with lentivirus containing three different sgRNA guides specific for the gene of interest. The lentiviruses were produced in HEK 293 T cell. Briefly, HEK293T cells were grown in 2 ml DMEM media

**Table 2.** Antibodies related to the experimental procedures. Antibodies used for immuno-blot and immuno-staining are listed below.

| Antigen | Company | RRID | Application |
|---|---|---|---|
| GABARAP | AbCam (Cambridge,UK) (Ab109364) | AB_10861928 | WB |
| GABARAP-L1 | Proteintech (Manchester, UK) (11010–1-AP) | AB_2294415 | WB, IF |
| FLAG | Sigma Aldrich (F7425-2MG) | AB_439687 | WB |
| FLAG | Sigma Aldrich (F1804) | AB_262044 | IF |
| HA | Roche (Mannheim, Germany) (11867423001) | AB_10094468 | IF |
| HA | Covance (Princeton, NJ, USA) (MMS-101P-1000) | AB_291259 | WB |
| LC3B | MBL (Woburn, MA, USA) (M152-3) | AB_1279144 | IF |
| LC3B | MBL (PM036) | AB_2274121 | IF |
| LC3B | CST (Danvers, MA, USA) (#2775) | AB_915950 | WB |
| RTN3 (human) | Bethyl (Montgomery, TX,USA) (A302-860A) | AB_10631136 | WB, IP |
| RTN3 | TebuBio (Offenbach am Main, Germany) (PA2256) | AB_2665372 | WB, IF |
| LAMP1 | DSHB (University of Iowa) (H4B4) | AB_528129 | IF |
| LAMP1 | DSHB (1D4B) | AB_2134500 | IF |
| LAMP1 | AbCam (Ab24170) | AB_775978 | IF |
| p62 | ENZO Life Science (Farmingdale, NY, USA) (PW9860) | AB_2196009 | WB |
| VINCULIN | Sigma-Aldrich (V9264) | AB_10603627 | WB |
| CALNEXIN | AbCam (Ab22595) | AB_2069006 | IF |
| REEP5 | Proteintech (14643–1-AP) | AB_2178440 | IF, WB |
| CLIMP-63 | Proteintech (16686–1-AP) | AB_2276275 | IF, WB |
| BSLC2 | AbCam (Ab106793) | AB_10974250 | IF |
| RTN1 | AbCam (Ab9274) | AB_307128 | WB |
| RTN4 | AbCam (Ab47085) | AB_881718 | WB |
| FAM134B | Gift from C. Hubner | Jena University | WB |
| ATG7 | CST (#8558) | AB_10831194 | WB |
| TRAP alpha | AbCam (Ab133238) | AB_11157579 | WB |
| KDEL | Merck-Millipore (10C3) | AB_212090 | IEM |
| CD63 | Ancell (Stillwater, MN, USA) (AHN16.1/46-4-5) | AB_2665375 | IEM |

*Table 2 continued on next page*

*Table 2 continued*

| Antigen | Company | RRID | Application |
|---------|---------|------|-------------|
| HA | Gift from G.Bu | Mayo Clinic, Jacksonville, FL,USA. | IEM |
| GFP | Santa Cruz (sc-9996) | AB_627695 | WB |

(10% FBS) in six-well plates until they reached 90% confluence. They were further co-transfecting with the three lentiviral plasmids (1.1 µg cDNA for each plasmid), containing the CAS9 and the selected sgRNA guides, together with the two packaging vectors pPAX2 (2.2 µg cDNA) and pMD2. G (1 µg cDNA). The medium containing lentivirus was collected after 24 hr and replaced with 2 ml of fresh DMEM subsequently collected after an additional 24 hr. The total 4 ml of DMEM media were pulled together and centrifuged to clear from dead HEK293T cells and stored at −80°C. One ml of the medium was then used to infect the desired cell lines and the rest was stored at −80°C. After 48 hr of infection, cells were selected using fresh DMEM media containing 5 µg/ml of Puromycin (further reduced to 2 µg/ml for the maintenance). When the pLenti-Puro-EGFP was used, cells were selected for GFP positivity through FACS sorting.

## Mass spectrometry

Interactome of FAM134B and RTN1-4 was performed using the SILAC labeling strategy. U2OS TRex cells transfected with mock plasmid were cultured in media supplemented with L-arginine-$^{12}C_6^{14}N_4$ (Arg0) and L-lysine-$^{12}C_6^{14}N_2$ (Lys0), while U2OS TRex expressing FAM134B or the various isoforms of the four RTNs were labeled with L-arginine-$^{13}C_6^{15}N_4$ (Arg10) and L-lysine-$^{13}C_6^{15}N_2$ (Lys8) as described previously (*Ong and Mann, 2006*; *Ong et al., 2002*). FAM134B and RTNs isoforms were induced for 24 hr with doxycycline and SILAC labeled cells were lysed using 50 mM Tris/HCl (pH 7.5), 120 mM NaCl, 1% (v/v) NP40, complete protease inhibitor cocktail (Roche). Cell lysates from light and heavy labeled cells were harvested separately and immunoprecipitated separately using monoclonal anti-HA-agarose antibody (Sigma). Only prior to proteins elution, heavy and light immuno-precipitation beads were combined together in a 1:1 ratio (H/L). Immunoprecipitated proteins were separated according to their molecular weight by subjecting them to SDS-PAGE electrophoresis. Gel lanes were cut into equal pieces and digested (*Shevchenko et al., 2006*). In brief, gel pieces were washed, de-stained and dehydrated. Proteins were reduced with 10 mM dithiothreitol (DTT), alkylated with 55 mM iodoacetamide (IAA) and digested with the endopeptidase sequencing-grade Trypsin (Promega; Madison, WI, USA) overnight at 37°C. Generated peptides were extracted using an increasing acetonitrile concentration. Collected peptide mixtures were concentrated and desalted using the Stop and Go Extraction (STAGE) technique (*Rappsilber et al., 2003*).

Instruments for LC-MS/MS analysis consisted of a NanoLC 1200 coupled via a nano-electrospray ionization source to the quadrupole-based Q Exactive HF benchtop mass spectrometer (*Michalski et al., 2011*). Peptide separation was carried out according to their hydrophobicity on an in-house packed 20 cm column with 1.9 mm C18 beads (Dr Maisch GmbH) using a binary buffer

**Table 3.** CRISPR-CAS9 guides sequences. CRISPR-CAS9 guide sequences used to generate KO cell lines in the manuscript are listed below.

| Gene | sgRNA sequence | Guide N° |
|------|---------------|----------|
| hATG7 | *CACC GAGAAGAAGCTGAACGAGTAT* | 1 |
| hATG7 | *CACC GCTGCCAGCTCGCTTAACA* | 2 |
| hATG7 | *CACC GTAAACTCTCTGGAAGACAGA* | 3 |
| hFAM134B | *CACC GATATCATTACATTTAAACAA* | 1 |
| hFAM134B | *CACC GCTTCCAGCTCAGCAGCTCGT* | 2 |
| hFAM134B | *CACC GCAATACAGTGGCTGAGCCT* | 3 |

system consisting of solution A: 0.1% formic acid (0.5% formic acid) and B: 80% acetonitrile, 0.1% formic acid (80% acetonitrile, 0.5% formic acid). 35 min gradients were used for immunoprecipitated samples. Linear gradients from 7 to 38% B in 20 min were applied with a following increase to 95% B within 5 min and a re-equilibration to 5% B. Q Exactive HF settings: MS spectra were acquired using 3E6 as an AGC target, a maximal injection time of 20 ms and a 60,000 resolution at 200 m/z. The mass spectrometer operated in a data dependent Top15 mode with subsequent acquisition of higher-energy collisional dissociation (HCD) fragmentation MS/MS spectra of the top 15 most intense peaks. Resolution for MS/MS spectra was set to 30,000 at 200 m/z, AGC target to 1E5, max injection time to 64 ms and the isolation window to 1.6 Th.

## GST pull-down

LC3s, GABARAPs and Ub proteins were cloned, as GST fusion proteins, into pGEX-4T-1 (GE Healthcare; Little Chalfont, UK) and expressed in Escherichia coli BL21 (DE3) cells in LB medium. Expression was induced by addition of 0.5 mM IPTG and cells were incubated at 37°C for 5 hr. Harvested cells were lysed using sonication in a lysis buffer (20 mM Tris-HCl pH 7.5, 10 mM EDTA, 5 mM EGTA, 150 mM NaCl) and GST fused proteins were immuno-precipitated using Glutathione Sepharose 4B beads (GE Healthcare). Fusion protein-bound beads were used directly in GST pull down assays. HEK293T cells were transfected with indicated constructs using GeneJuice (Merck) for 24 hr. Cells were lysed in lysis buffer [50 mM HEPES, pH 7.5, 150 mM NaCl, 1 mM EDTA, 1 mM EGTA, 1% Triton X-100, 10% glycerol supplemented with protease inhibitors (Complete, Roche). SH-SY5Y cells were used for endogenous pull down of RTN3. Lysates were cleared by centrifugation at 12,000 g for 10 min, and incubated with GST fusion protein-loaded beads over-night at 4°C. Beads were then washed three times in lysis buffer, resuspended in Laemmli buffer and boiled. Supernatants were loaded on SDS-PAGE.

## Fluorescence microscopy

Cells were plated on glass cover slips, and 24-hr post induction, with 1 µg/ml doxycycline, they were fixed by 4% (wt/vol) paraformaldehyde for 5 min. Cells were permeabilized with a 0.1% Saponin solution in PBS at room temperature for 5 min and blocked in PBS containing 10% fetal bovine serum (FBS) for 1 hr at room temperature. Cells were incubated overnight at 4°C with primary antibody diluted in PBS containing 5% FBS and 0.1% Saponin. Washes were performed in 0.1% Saponin in PBS. Secondary antibodies were incubated for 1 hr at room temperature and washed two times with PBS 5% FBS, 0.1% Saponin and once with PBS. The coverslips were mounted on 10 µl of aqueous mounting medium (Mowiol) and placed on a glass holder. Images were acquired with the Leica SP8 laser-scanning microscope (Leica) or with CSUX1 Real-Time Confocal System with Nipkow spinning disk (Visitron). Images shown are representative of experiments carried out at least three times.

## Dimerization/oligomerization assay

U2OS TRex cells were plated on glass cover slips. Cells were co-transfected with 0.5 µg of pC4-RhE-FRB (T2098L)-FLAG-RTN3B plasmid together with 0.5 µg of one of the following: pC4M-F2E-FKBP-HA-RTN3B or pC4M-F2E-FKBP-HA-RTN3A. After 24 hr from transfection, cells were treated with 500 ng Rapalog in DMEM, 10% FBS for 2 hr and then fixed in 4% PFA. Samples were immunostained, as previously described, with primary antibodies against FLAG, HA, LAMP1 and LC3B. Single transfections with pC4-RhE-FRB (T2098L)-FLAG-RTN3B, pC4M-F2E-FKBP-HA-RTN3B, pC4M-F2E-FKBP-HA-RTN3A, were performed as internal controls.

## Super-resolution microscopy

Cells were plated on glass cover slips, and 24 hr post induction, with doxycycline, they were fixed by 4% (wt/vol) paraformaldehyde for 5 min. Cells were permeabilized with a 0.1% Saponin solution in PBS at room temperature for 5 min and blocked in PBS containing 10% fetal bovine serum (FBS) for 1 hr at room temperature. Cells were incubated overnight at 4°C with primary antibody diluted in PBS containing 5% FBS and 0.1% Saponin. Washes were performed in 0.1% Saponin in PBS. Secondary antibodies were incubated for 1 hr at room temperature and washed two times with PBS 5% FBS, 0.1% Saponin and once with PBS. Samples were fixed with 2% PFA (wt/vol) for 2 min at room temperature, washed in PBS and stored at 4°C in PBS solution. Samples were exposed to a solution

of 1:700 TetraSpeck microsperes (0.1 μm, Thermo Fisher) in PBS for 10 min, and subsequently rinsed with PBS. Cells were imaged in reducing buffer that facilitated photo-switching for dSTORM (Heilemann et al., 2008). The reducing buffer contained 100 mM MEA (Sigma) in PBS with pH adjusted to 8.0–8.5 using KOH from 1 M stock solution, and was freshly prepared before each experiment. Single-molecule imaging was performed on a custom-built wide-field microscope as described in Dietz et al., 2013. A 643 nm (iBeam smart, Toptica Photonics), and a 532 nm (DPPS-532-NL300, Eksma Optics, Lithuania) laser were combined by appropriate dichroic mirrors (Laser-MUX 561–594R, LaserMUX 514–543R, LaserMUX 473–491R, 1064R, LaserMUX 427–25, AHF) and coupled into an inverted microscope (IX71, Olympus) equipped with a 100× oil immersion objective (PLAPO 100× TIRFM, NA ≥1.45, Olympus) and a nose piece (IX2-NPS, Olympus) to avoid drift. A translational mirror (Thorlabs) within the illumination pathway allowed for a continuous adjustment of the illumination light field angle from wide-field to total internal reflection (TIR) illumination. The 643 and 532 nm lasers were addressed separately by application of an acousto-optic tunable filter (AAOptics). Fluorescent light was collected by the objective and spectrally separated from excitation light by a dichroic mirror (Dualline zt532/638rpc, AHF) inside the microscope and bandpass filters (ET 700/75 for Alexa Fluor 647 and ET 575/50 for Alexa Fluor 532, both from AHF) situated in the emission pathway. Filtered fluorescent light was projected on an EMCCD (iXon3, Andor). Dual-color microscopy was performed with the EMCCD camera operating in frame transfer video mode at a 200-fold electron multiplying gain. Imaging was conducted sequentially by imaging longer wave-length absorption first, that is, Alexa Fluor 647 prior to Alexa Fluor 532. After TIR illumination was established, Alexa Fluor 647 was read out by recording image sequences of about 20,000 images with a frame rate of 33 Hz under continuous 643 nm laser illumination (2 kW/cm$^2$). Lasers and filters were then rearranged for photo-activating and detecting Alexa Fluor 532. Image sequences of about 20,000 images were recorded with a frame rate of 10 Hz under constant 532 nm laser excitation (3 kW/cm$^2$). SMLM data were analyzed by calculating the fluorescent label positions by fitting their point spread functions (PSFs) from single-molecule emissions to a two-dimensional Gaussian intensity distribution using rapidSTORM (Wolter et al., 2010). A local relative threshold of 100 and of 150 was set for Alexa Fluor 647 and Alexa Fluor 532, respectively, to omit dim emitters. Coordinates of fluorescent labels were stored in charts, further referred to as localization lists. Based on the localization lists, images were reconstructed. TetraSpeck microspheres were used for spatial alignment of dual-color images.

## Ultrastructural analyses

For the IEM analyses, U2OS RTN3L cells were starved in EBBS for 6 hr in presence of 200 nM bafilo-mycinA1 before being fixed by addition of 4% PFA and 0.4% glutaraldehyde in 0.1 M phosphate buffer (pH 7.4) in an equal volume to the DMEM medium and incubated for 20 min at room temperature. Cells were subsequently fixed in 2% PFA and 0.2% glutaraldehyde in 0.1 M phosphate buffer (pH 7.4) for 3 hr at room temperature and then embedded for the Tokuyasu procedure before cutting ultrathin cryo-sectioning and immuno-gold label them, as previously described (Slot and Geuze, 2007). Primary antibodies used were against the KDEL, HA protein tag and CD63. The first two labeling were revealed with protein A conjugated with 10 nm gold and the second one with CD63 protein A conjugated with 15 nm gold. Cell sections were finally analysed using an 80KV transmission electron microscope (Jeol 1200-EX).

## Immunoblotting

Protein lysates were resolved in SDS-PAGE gels (8%–10–12% Bisacrylamid gels or 4–20% gradient gels [BioRad; Hercules, CA, USA]) and transferred to nitrocellulose (GVS North America, 0.45 μm; Stanford, ME, USA) or PVDF (Merck-Millipore, 0.2 μm) membrane. Membranes were blocked in TBS 0.1% Tween containing 5% low fat milk (Roth; Karlsruhe, Germany) and incubated overnight at 4°C with the specific primary antibody. Immuno-blot bands were quantified using ImageJ program.

## Co-immunoprecipitation

Stable cell lines for long and short isoforms of RTNs were induced with doxycycline (1 μg/ml) for 24 hr. A confluent 10 cm diameter petri dishes U2OS TRex cells expressing the desired isoform of the RTNs was harvested in 50 mM Tris/HCl (pH 8), 120 mM NaCl, 1% (v/v) NP40, complete protease

inhibitor cocktail (Roche), and 1 mM PMSF. Lysates were cleared by centrifugation at 12,000 g for 10 min, and incubated overnight at 4°C with monoclonal anti-HA-agarose antibody (Sigma Aldrich). Beads were then washed three times in lysis buffer, suspended in Laemmli buffer and boiled. Supernatants were loaded on SDS-PAGE.

HEK 293T cells were co-transfected with the indicated constructs using GeneJuice (Merck) for 24 hr. Cells were lysed in lysis buffer [50 mM HEPES, pH 7.5, 150 mM NaCl, 1 mM EDTA, 1 mM EGTA, 1% Triton X-100, 10% glycerol, 25 mM NaF supplemented with PMSF (Sigma Aldrich), aprotinin (Sigma Aldrich), leupeptin (Biomol; Hamburg, Germany), and sodium vanadate (Sigma Aldrich)]. Lysates were cleared by centrifugation at 12,000 g for 10 min, and incubated overnight at 4°C with monoclonal anti-HA-agarose antibody (Sigma Aldrich) or 2 hr at 4°C with GFP-TRAP beads (Chromoteck; Planegg, Germany). Beads were then washed three times in lysis buffer, resuspended in Laemmli buffer and boiled. Supernatants were loaded on SDS-PAGE.

For endogenous co-immunoprecipitation between RTN3 and GABARAP, a confluent 15 cm diameter petri dishes of A549 cells was harvested in 50 mM Tris/HCl (pH 8), 120 mM NaCl, 1% (v/v) NP40, complete protease inhibitor cocktail (Roche), and 1 mM PMSF. Lysates were cleared by centrifugation at 12,000 g for 10 min, and incubated overnight at 4°C with RTN3 primary antibody. Lysates and RTN3 antibody solutions were incubated with Protein A-Agarose beads (Roche) for 4 hr at 4°C. Beads were then washed three times in lysis buffer, resuspended in Laemmli buffer and boiled. Supernatants were loaded on SDS-PAGE.

## Data analysis

All experiments were performed in at least three independent biological replicates. For cell microscopy, (unless differently specified in the figure legend) a minimum number of 100 cells, for each condition, were considered for data quantification. Data are presented as mean ± s.d. Statistical analysis of two experimental groups was performed using parametric two-tailed Student's $t$ test.

For mass spectrometry, all acquired raw files were processed using MaxQuant (1.5.3.30) and the implemented Andromeda search engine. For protein assignment, spectra were correlated with the Uniprot human database (v. 2016) including a list of common contaminants. Searches were performed with tryptic specifications and default settings for mass tolerances for MS and MS/MS spectra. Carbamidomethyl at cysteine residues was set as a fixed modification, while oxidations at methionine, acetylation at the N-terminus were defined as variable modifications. The minimal peptide length was set to seven amino acids, and the false discovery rate for proteins and peptide-spectrum matches to 1%. The match-between-run feature with a time window of 1 min was used. For further analysis, the Perseus software (1.5.3.0) was used and first filtered for contaminants and reverse entries as well as proteins that were only identified by a modified peptide. The SILAC Ratios were logarithmized and grouped into duplicates and a one-sample t-test was performed. Probability values (p) < 0.05 were considered statistically significant. To identify significant enriched GO terms, we utilized the 1D enrichment tool in Perseus. Data visualization was done in the statistical environment R. MS analyses of three independent immuno-precipitations (IP) experiments for both isoforms of each RTN were performed. Peptides with $Log_2$ (Heavy/Light [H/L]) ratios $\geq 1$ and a p value $\leq 0.05$ were considered significantly enriched. Pearson's correlation coefficients up to 0.9 indicated the high reproducibility between biological repetitions.

## Acknowledgements

We would like to thank Manuel Kaulich, Christian Behrends, Hesso Fahran and Richard Youle for providing us vectors and cell lines; Prof. Noboru Mizushima for $Atg5^{-/-}$ MEFs; Prof. Jun-Lin Guan for $Fip200^{-/-}$ MEFs, Prof. Christian Hübner for $Fam134b^{-/-}$ MEFs, Prof. Stephen Blacklow for U2OS TRex cells, Prof. S. Taylor for HeLa TRex cells. We thank Viktória Major and Laura Pisano for helping in the experimental work. We acknowledge Alexandra Stolz, Daniela Hoeller and Anna Vainshtein for critical reading of the manuscript and valuable insights. This work was supported by grants from the DFG (SFB 1177) and the Cluster of Excellence 'Macromolecular Complexes' of the Goethe University Frankfurt (EXC 115) to ID, MH, SM; LOEWE grant Ub-Net to ID, SM and LOEWE Centrum for Gene and Cell therapy Frankfurt (CGT) to ID PG was supported by the 7.FP, COFUND, Goethe International Postdoc Programme GO-IN, No. 291776. FR is supported by SNF Sinergia (CRSII#_154421), ZonMW VICI (016.130.606) and Marie Sklodowska-Curie Cofund grants.

## Additional information

### Competing interests

ID: Senior editor, *eLife*. The other authors declare that no competing interests exist.

### Funding

| Funder | Grant reference number | Author |
| --- | --- | --- |
| Deutsche Forschungsgemeinschaft | Collaborative Research Center on Selective Autophagy SFB1177 | Stefan Müller<br>Mike Heilemann<br>Ivan Dikic |
| Cluster of Excellence Goethe University Frankfurt am Main | EXC 115 | Mike Heilemann<br>Ivan Dikic |
| LOEWE programme | Network Ub-Net | Stefan Müller<br>Ivan Dikic |
| LOEWE Center for Gene and Cell Therapy Frankfurt | CGT | Ivan Dikic |
| 7.FP, COFUND, Goethe International Postdoc Program GO-IN | No 291776 | Paolo Grumati |
| SNF Sinergia | CRSII#_154421 | Fulvio Reggiori |
| ZonMw | VICI (016.130.606) | Fulvio Reggiori |
| Marie Sklodowska-Curie Cofund | | Fulvio Reggiori |

The funders had no role in study design, data collection and interpretation, or the decision to submit the work for publication.

### Author contributions

PG, Conceptualization, Data curation, Formal analysis, Investigation, Methodology, Writing—original draft, Project administration, Writing—review and editing; GM, Data curation, Formal analysis, Investigation, Writing—review and editing; SH, Data curation, Formal analysis, Methodology, Writing—review and editing; MM, M-LIEH, Formal analysis, Investigation, Writing—review and editing; RY, Resources; SM, Writing—original draft, Writing—review and editing; FR, MH, Supervision, Writing—review and editing; ID, Conceptualization, Data curation, Supervision, Funding acquisition, Methodology, Writing—original draft, Project administration, Writing—review and editing

### Author ORCIDs

Ivan Dikic, http://orcid.org/0000-0001-8156-9511

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
