## [Decision Letter]

Thank you for submitting your article "RTN3 regulates tubular endoplasmic reticulum turnover via selective autophagy" for consideration by *eLife*. Your article has been reviewed by three peer reviewers, and the evaluation has been overseen by a Reviewing Editor and Randy Schekman as the Senior Editor. The reviewers have opted to remain anonymous.

The reviewers have discussed the reviews with one another and the Reviewing Editor has drafted this decision to help you prepare a revised submission.

Summary:

The present study by Grumati et al. characterizes the role of selective autophagy in the maintenance of the ER membrane. The authors identified RTN3L as an important new player in starvation-induced fragmentation of tubular ER membranes coupling this process with the recruitment of the autophagic machinery to promote ERphagy. RTN3L is therefore proposed to act as an autophagic receptor for ERphagy acting by interacting with different Atg8 family members in a LIR-dependent manner. Overall, this study tackles and important question in cell biology in general and in autophagy in particular. The data characterizing RTN3L, as a novel autophagic receptor, is convincing and the manuscript is clearly presented. However, the reviewers found that additional work is required to strengthen the manuscript to meet the standards for *eLife*.

Essential revisions:

1) This study relies on only exogenously overexpressed RTN3L. To prove that RTN3L mediates autophagic degradation of the ER as shown in Figure 8, it is essential to test whether ER degradation is indeed affected in RTN3L KO or KD cells expressing the RTN3LΔ6LIR mutant or RTN3LΔ6LIR knock-in cells.

2) The authors only see degradation of overexpressed RTN3L itself. It does not represent ERphagy. In order to mention "ERphagy", it is essential to determine lysosomal degradation of ER resident proteins using WB and IF approaches.

3) It is unclear what the RTN3L punctate structures represent. Are they fragmented ER, ER fragments in autophagosomes, or autolysosomes? This should be addressed by electron microscopy and/or density gradient analysis.

4) Related to comment #3, "fragmentation of the ER" and "sequestration of the ER into autophagosomes" may occur sequentially, but these should be distinct processes. Does RTN3L mediate both steps?

5) Given its localization to the tubular ER, the authors claim that RTN3L mediates degradation of this specific subdomain of the ER. However, it seems that experimental evidence for this proposal has not been provided. It should be investigated whether proteins in the tubular ER is preferentially degraded by RTN3L-driven ER-phagy compared with those in the other ER subdomains.

---

## [Author Response]

*Essential revisions:*

*1) This study relies on only exogenously overexpressed RTN3L. To prove that RTN3L mediates autophagic degradation of the ER as shown in Figure 8, it is essential to test whether ER degradation is indeed affected in RTN3L KO or KD cells expressing the RTN3LΔ6LIR mutant or RTN3LΔ6LIR knock-in cells.*

In order to address this point, we performed several experiments using *Rtn3* knockout MEFs. The lack of Rtn3 blocked the degradation of several ER proteins, which are mainly resident in ER tubules. Of note, ER sheets proteins turnover was not compromised and also the global macro-autophagy flux was comparable to the one observed in wild-type cells. Data are presented in Figure 4 and Figure 4—figure supplement 1.

The normal ER tubules proteins turnover was rescued in Rtn3 knockout MEFs after reconstitution with RTN3L but not with RTN3LΔ6LIRs or the short isoform RTN3S (Figure 4 and Figure 7). The old Figure 8 is now Figure 9.

*2) The authors only see degradation of overexpressed RTN3L itself. It does not represent ERphagy. In order to mention "ERphagy", it is essential to determine lysosomal degradation of ER resident proteins using WB and IF approaches.*

In order to address this point, we performed new IF experiments where we showed that several ER resident proteins (CALNEXIN, CLIMP-63, REEP5 and RTN3) are degraded via lysosomes after EBSS treatment in cells were none of the RTNs were expressed (Figure 3—figure supplement 1). The same results were confirmed via WB analysis, where we showed that Rtn1, Rtn3, Rtn4, Reep1, Trap α, Climp63, Fam134b were degraded via lysosomes after nutrient deprivation (Figure 4; Figure 4—figure supplement 1). Moreover, in cells over-expressing RTN3L, together with RTN3L were degraded other ER proteins like CALNEXIN, BSCL2 and the ER tubules protein REEP5 (Figure 3—figure supplement 1).

*3) It is unclear what the RTN3L punctate structures represent. Are they fragmented ER, ER fragments in autophagosomes, or autolysosomes? This should be addressed by electron microscopy and/or density gradient analysis.*

To address this point, we performed immuno-electron-microscopy analysis on U2OS HA-FLAG-RTN3L cells after EBSS treatment. In details, we performed the following labeling:

anti-HA to highlight RTN3L presence inside autophagosomes and autolysosomes;

anti-KDEL to highlight ER structure inside autolysosomes;

anti-HA combined with anti-CD63, a late endo-lysosomal membrane protein marker;

anti-KDEL combined with anti-CD63-biotin.

In U2OS RTN3L cells treated with EBSS, ER fragments labelled for KDEL are often found inside autolysosomes (AL) as shown in Figure 3—figure supplement 5. This phenotype was not observed when cells were grown in normal media (data not shown). HA labelling (to visualize RTN3L) was also detected inside autophagosomes and autolysosomes in U2OS-RTN3L cells after EBSS treatment (Figure 3—figure supplement 5). The double labelling KDEL–CD63 and HA(RTN3L)–CD63 confirmed the presence of RTN3L-positive vesiculo-tubular structures inside autolysosomes (Figure 3).

*4) Related to comment #3, "fragmentation of the ER" and "sequestration of the ER into autophagosomes" may occur sequentially, but these should be distinct processes. Does RTN3L mediate both steps?*

We believe that RTN3L is able to mediate both processes. RTN3L dimerization is functional and sufficient to fragment ER tubules as we showed using the FKBP-RTN3L – FBP-RTN3L co-transfection in combination with Rapalog treatment. The forced dimerization of RTN3L was sufficient to induce ER tubules fragmentation when cells were kept in standard growing conditions (DMEM with 10% FBS), and the Rapalog treatment itself did not affect the basal autophagy flux. However, autophagy is required to catalyze the fragmentation process. When macro-autophagy was impaired, RTN3L ability to fragment ER tubules was compromised. Moreover, the presence of the LIR motives is than necessary to attract the autophagy machinery and to recruit the LC3s modifiers in order to form the autophagosome and promote the subsequent delivery to the lysosomes. The mutant form RTN3LΔ6LIR was, in fact, unable to fragment ER tubules.

*5) Given its localization to the tubular ER, the authors claim that RTN3L mediates degradation of this specific subdomain of the ER. However, it seems that experimental evidence for this proposal has not been provided. It should be investigated whether proteins in the tubular ER is preferentially degraded by RTN3L-driven ER-phagy compared with those in the other ER subdomains.*

In order to address this point, we analyzed how different ER proteins were degraded in *Rtn3* vs *Fam134b* knockout MEFs. We previously demonstrated that Fam134b is a ER-phagy receptor mainly dedicated to ER sheets turnover (Khaminets et al. Nature 2015).

ER tubules resident proteins like RTN1 and REEP5 were more stable in *Rtn3* knockout MEFs respect to ER sheets proteins like Climp-63 and Trap-α, after EBSS starvation. Contrary, in *Fam134b* knockout cells, ER tubules proteins were normally degraded while ER sheets proteins were more stable (Figure 4 and Figure 4—figure supplement 1).

Moreover, CLIMP-63 never co-localized with the ER-tubules fragments positive for RTN3L; neither after RTN3L over-expression followed by 6h EBSS starvation (Figure 3—figure supplement 3) nor after Rapalog induced dimerization of FKBP-RTN3L and FBP-RTN3L co-overexpression (Figure 2—figure supplement 2). Contrary, REEP5 always co-localized with RTN3L positive structures (Figure 3—figure supplement 3; Figure 2—figure supplement 2).